# An assessment of China's methane mitigation potential and costs and uncertainties through 2060

Nina Khanna [1], Jiang Lin [1,2] ✉, Xu Liu [3] & Wenjun Wang[2]

China, the world's largest methane emitter, is increasingly focused on methane mitigation in support of its climate goals, but gaps exist in the understanding of key methane sources, as well as mitigation opportunities and their associated uncertainties. We use a bottom-up modeling approach with updated methane emission projections and abatement cost analysis to account for additional sources, uncertainties, and mitigation measures in China's energy and agricultural sectors. Here we show the significant cost-effective potential for reducing methane emissions in China by 2030, with 660 million tonnes of carbon dioxide equivalent possible with average negative abatement costs of US\$6.40 per tonne $CO_2e$. Most of this potential exists in the energy sector, particularly coal mining, but the greater potential will shift towards agriculture by 2060. Aquaculture and biochar applications in rice cultivation have net economic benefits but need greater support for deployment, while new mitigation measures will be needed for remaining emissions from enteric fermentation, rice cultivation, and wastewater.

There is increasing recognition from the scientific community that limiting global temperature increase to below 1.5 °C requires not only limiting cumulative carbon dioxide ($CO_2$) emissions, but also strong reductions in other non-$CO_2$ greenhouse gases (GHGs)[1,2]. Emissions of methane, a short-lived climate pollutant and potent greenhouse gas, have contributed to approximately 30% of the current rise in global average temperatures[3,4]. Reducing methane emissions is critical to slowing the adverse impacts of climate change in the near term and could help avoid nearly 0.3 °C of global temperature increase by the 2040s[3]. Although methane has a shorter atmospheric lifetime than $CO_2$ (12 years vs 100 years), its global warming impact is up to 87 times greater than $CO_2$ over a 20-year timeframe (GWP20), and up to 36 times greater over a 100-year timeframe (GWP100)[5]. By 2030, measures that reduce methane can cut warming more significantly than those targeting only $CO_2$ emissions, due in part to reductions of co-emitted pollutant aerosol particles from fossil fuel combustion that help cool the planet[6]. Additionally, because methane emissions contribute to ozone pollution and accompanying adverse effects on human health and agricultural productivity, its reduction can provide co-benefits in improved air quality, better health conditions, and increased crop yields[3].

China is the world's largest methane emitter, accounting for nearly one-fifth of total global methane emissions[3]. China's national climate change policies did not focus significantly on methane until the early 2010s, although prior Clean Development Mechanism projects had included methane mitigation[7]. More recently, China's focus on the mitigation of anthropogenic methane emissions as part of its climate change strategy has been elevated in both its domestic policies and international commitments. Domestically, China's 14th Five-Year Plan for National Economic Development, endorsed in 2021, explicitly included text that China will "strengthen the control" of non-$CO_2$ GHGs including methane for the first time. On November 7, 2023, China released its national methane emissions control action plan and later committed to include methane and other non-$CO_2$ GHGs in its forthcoming Nationally Determined Commitment for 2035[8,9]. The national methane action plan prioritizes significantly enhancing methane monitoring, reporting, and verification systems, and calls for effectively improving methane utilization, emissions control technologies,

[1]Energy Technologies Area, Lawrence Berkeley National Laboratory, Berkeley, CA, USA. [2]University of California at Berkeley, Berkeley, CA, USA. [3]Peking University, Beijing, China. ✉e-mail: j_lin@lbl.gov

and policy frameworks in the energy, agriculture, and waste sectors[8]. However, the action plan did not set quantitative targets on methane emissions control, and only included four quantitative goals for increasing the utilization of coal mine gas, reutilization of livestock waste and urban household waste, and harmless disposal of urban sludge. At the sectoral level, controls on methane emissions have been qualitatively discussed in other domestic sector plans, such as actions to control and reduce coal capacity, minimize household waste, improve agricultural management, and increase gas recovery and recycling in oil and gas production but quantitative targets continue to be lacking[10].

As a non-Annex I country, China has reported its GHG emissions, including methane, in its national communication reports submitted to the United Nations Framework Convention on Climate Change in 2004, 2012, 2019, and 2023, but its national GHG emissions inventory only covers emissions data up to 2017[11]. China follows the 2006 Intergovernmental Panel on Climate Change (IPCC) Guidelines for preparing its national inventory, and currently follows a multi-institutional approach to compiling the inventory data from various relevant ministries, industry associations, and research and academic organizations led by the Ministry of Ecology and Environment[12]. As methane is a relatively new area in China's climate change mitigation and policy development efforts, a strong scientific and analytical basis for understanding the country's key methane emission sources and mitigation opportunities, along with their associated uncertainties, is critical to developing an effective methane mitigation roadmap.

Recent literature and analysis of China's anthropogenic methane emissions and mitigation potential have mostly focused on specific sectors, such as coal mining[4,13–17] and agriculture[18–20] and, less commonly, solid waste[21,22]. These studies explore sector-specific data sources, challenges, and mitigation opportunities in detail but do not provide a comprehensive national view of methane emissions and opportunities, and only a few attempt to quantify areas of uncertainty. We address this knowledge gap by exploring what are the under-reported anthropogenic methane emission sources in China and what are key sources of uncertainties for key emission sources such as coal mine methane.

Other recent studies[7,23,24] have included multi-sector modeling and analysis of China's methane sources and mitigation potential. EPA[23] and Lin et al.[24] both use a bottom-up approach to projecting China's methane emissions under different mitigation scenarios, while Yu et al.[25] review estimated historical emissions from top-down and bottom-up inventories, and projections from four different energy system models. Most of these projections focus on 2050 as the end year for their analysis, rather than 2060—the target year of China's carbon neutrality target—and use dated input parameters prior to 2020. We build on this to address how China's methane emissions sources change over time through 2060, taking into consideration updated projections of key emission drivers such as population, coal production, and updated mitigation costs data. We further explore what are new and emerging opportunities for methane mitigation potential for each emission source and their related costs, given technical feasibility and current cost estimates.

In addressing the existing research gaps, this paper provides an analysis using a bottom-up modeling approach with updated assumptions about macroeconomic and physical emissions drivers that extend methane emissions and mitigation projections through 2060 using China-specific abatement cost data. We use a business-as-usual Reference Scenario without methane mitigation as the baseline scenario, and two methane mitigation scenarios (defined by current abatement cost thresholds of <US$10/tCO$_2$e for Cost-effective Mitigation, and <US$100/tCO$_2$e for Deep Mitigation for considering individual mitigation measures), we comprehensively assess the mitigation potential for individual methane sources in both the near term (through 2030) and the long term (through 2060). Through

bottom-up modeling and scenario analysis, we contribute to the understanding of new and emerging opportunities to reduce China's methane emissions cost-effectively in three important ways. First, we account for additional methane emission sources, such as abandoned coal mines and aquaculture, that have not yet been incorporated in multi-sectoral analyses of China. We also address uncertainty in methane emissions data in two specific ways: by using more granular, region-weighted emission factors for the coal mining sector, and by assessing uncertainties in specific rice cultivation mitigation measures. Lastly, we evaluate new and emerging mitigation opportunities, such as biochar application in rice cultivation and aquaculture. We identify significant cost-effective potential for reducing methane emissions in China by 2030, with 660 MtCO$_2$e reductions possible with average negative abatement costs of US$6.40/tCO$_2$e. In the longer term to 2060, the agriculture sector holds greater mitigation potential but will need policy support such as government funding in research, development and deployment, and inclusion in China's voluntary carbon credit market to come to fruition.

## Results

There are generally two methods for estimating methane emissions in compiling emissions inventories: bottom-up and top-down. Bottom-up methods such as point-source measurements and facility-scale in situ aircraft measurements help improve the understanding of the process of emissions generation and the development of possible mitigation strategies. Top-down methods monitor the spatial and temporal trends of emissions through the use of remote observatories, towers, and satellites. The two types of methods are complementary, and top-down results help enable the rigorous comparison with bottom-up results[26]. For methane emissions specifically, top-down measurement initiatives have only recently emerged to help improve the measurement and reporting of current and historical emissions, but these assessments remain incomplete and most countries—including China—have little or no measurement-based data[4]. Regardless of the method, large uncertainties exist in methane emissions inventories, and they come from various sources. For example, uncertainties from bottom-up methods may relate to uncertainties in activity-level data due to incompleteness or lack of representativeness of statistical sampling, as well as the imputation of missing data and extrapolation for future years. Additionally, uncertainties in emission factors (EFs) may arise due to the representativeness of a limited number of observations, inaccuracies in assumptions and/or source aggregation, as well as biases, variability, and/or random errors[27]. In contrast, uncertainties from top-down methods may relate to instrumentation precision and inverted atmospheric transport model assumptions. Estimating the level of uncertainty is essential since it reveals both the accuracy and confidence level of emissions and inventory estimates. In the absence of robust measurement-based data for multiple sectors of China, this study uses a standardized bottom-up method based on existing international guidelines for greenhouse gas emission inventories to estimate the historical and projected China's future methane emissions.

### Current methane emissions in China and uncertainties

Based on updated information and analysis about the source activities that drive methane emissions ("source activity drivers"), we estimate that China's 2020 anthropogenic methane emissions totaled 63.8 million metric tonnes (MtCH$_4$), which is equal to 1913 million metric tonnes of carbon dioxide equivalent (MtCO$_2$e) having a 100-year global warming potential (GWP100). Emissions sources include 41% from agriculture, 46% from energy, and 13% from waste and wastewater. Figure 1 compares our total methane emissions estimates for 2020 with other recently published estimates that are primarily based on bottom-up inventory approaches. Of the studies shown in Fig. 1, only IEA's estimate[4] attempts to update their bottom-

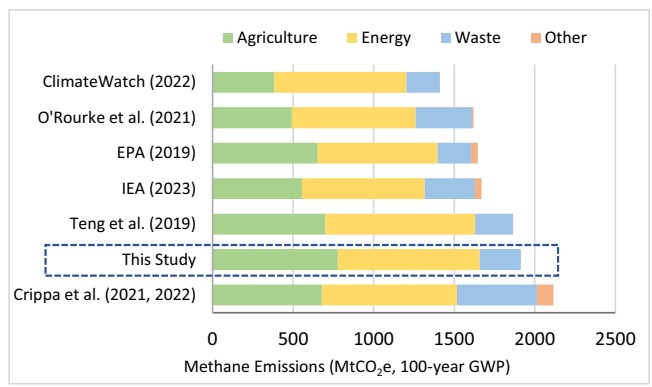

**Fig. 1 | Estimates of China's 2020 methane emissions by sector.** Comparison of emission results from this study shown in the dashed box with other recent estimates from ClimateWatch[49], O'Rourke et al.[50], U.S. EPA[23], IEA[4], Teng et al.[38], and Crippa et al.[28] Other category shown varies by different data sources and includes chemicals, metals and fossil fuel fires in ref. 50, manufacturing, other transport, chemical, and metal industries, and fires in ref. 28, stationary and mobile sources in EPA[23], and an average of other estimates from four key sources in IEA[4]. Teng et al.[38] estimate is based on its BAU scenario. Source data are provided as a Source Data file.

up estimates by incorporating findings from satellite measurement campaigns and measurement-based, peer-reviewed studies. Variations in estimates for the agriculture and waste sectors indicate greater uncertainty about source activity data and emission factors in both sectors. This is particularly true for the EDGAR global emissions inventory[28], which primarily uses the Intergovernmental Panel on Climate Change (IPCC) Tier 1 global default data for emission factors with greater inherent uncertainties than region-specific data for these two sectors. In the agriculture sector, for example, enteric fermentation and manure management had an estimated uncertainty of ±20% for activity data and ±50% in Tier 1 emission factors, with higher uncertainty levels of −40% to +70% for the emission factor for rice cultivation[27]. In the waste sector, similarly, high uncertainty levels of −56% to +103% are seen in activity data for industrial wastewater, with 30% uncertainty in maximum methane-producing capacity and −50% to +100% uncertainty in methane correction factor, all of which are input variables that directly impact the emission factor[27].

Our estimate of China's methane emissions falls on the higher end of such estimates, which is likely due to the inclusion of two additional methane sources: abandoned coal mines and aquaculture. We also used regionally weighted median emission factors for coal mine methane, which enabled us to reduce some uncertainty in the energy sector. In terms of sectoral composition, our estimate of agricultural methane emissions (including 16% from aquaculture) is larger than other studies, but our estimates of the energy and waste sectors are comparable.

### Future methane trajectories and uncertainties

In China, key sources of uncertainty identified across bottom-up and top-down inventory estimates for 2017 include coal mining, rice cultivation, wastewater, and enteric fermentation[7]. As discussed earlier, we focused on coal mining and rice cultivation as two large sources of methane emissions and conducted additional uncertainty analysis around emission factors and mitigation efficacy. Figure 2a–c shows projected methane emissions and associated uncertainties under three scenarios: Reference, Cost-effective Mitigation, and Deep Mitigation. Because mitigation measures are assumed to be deployed from 2020 onwards under the two mitigation scenarios, total methane emissions begin declining after 2020 under both the Cost-effective and Deep Mitigation scenarios, but do not decline until after 2025 under the Reference scenario with no mitigation measures.

The main sources of uncertainties shown in Fig. 2a are coal mine methane and rice cultivation, hence our focus on those areas. Uncertainties about the coal mine methane emission factor in the range of ±80% are similar to the range of other global uncertainty estimates of > ±100%[27]. The significantly lower uncertainty range of ±10% identified for rice cultivation mitigation efficiency is lower than that for global methane emission factors for rice cultivation in the existing EDGAR emissions inventory, as we only considered the uncertainty in efficacy of biochar specific to China and not the inherent uncertainties in emission factors[27]. For coal mine methane, the overall uncertainty level falls over time as the economy shifts away from coal consumption and coal mining decreases until it is nearly phased out by 2060 (Fig. 2b). For rice cultivation, overall uncertainty levels in both the Cost-effective and Deep Mitigation scenarios are low but remain constant due to uncertainty about the efficacy of a mitigation measure—namely, the application of biochar (i.e., partially combusted biomass) as a soil supplement to absorb methane—which is fully deployed by 2030 (Fig. 2c). Despite these uncertainties, a median reduction in total methane emissions of 58% below the reference scenario is possible by 2060 under the Deep Mitigation scenario.

Figure 3 compares the annual methane reduction potential by sector over time under both the Cost-Effective (Fig. 3a) and Deep Mitigation (Fig. 3b) scenarios. Under both scenarios, most reduction potential will be from the energy sector—notably from coal mining in the earlier years (2020–2040) and abandoned coal mines in the later years (2025–2060). In the later years and particularly after 2050, coal mining reduction potential decreases due to both activity decreases as China shifts away from coal production and emissions control as China fully deploys measures that destroy or utilize ventilated air methane (VAM) released from coal mines by 2050. Under the Deep Mitigation scenario, a slightly greater reduction potential exists on an accelerated timeframe (through 2040) due to faster deployment of VAM mitigation measures (by 2030, rather than 2050) and faster phasedown of coal consumption in an accelerated clean energy transition. In the agricultural sector, the greater methane reduction potential seen under the Deep Mitigation scenario hinges on the deployment of additional mitigation measures in rice cultivation: in 2030, the application of changes to irrigation practices reduces methane emissions by an additional 47 $MtCO_2e$ compared to the Cost-Effective Scenario. By 2060, these irrigation changes result in a total reduction potential of 317 $MtCO_2e$ under the Deep Mitigation scenario, and 269 $MtCO_2e$ under the Cost-effective Mitigation Scenario. Similarly, the implementation of higher cost mitigation measures (e.g., anaerobic treatment with gas recovery and utilization and aerobic wastewater treatment) under the Deep Mitigation scenario results in 22 $MtCO_2e$ and 28 $MtCO_2e$ of methane reduction potential in 2030 and 2060, respectively.

In 2030, most methane mitigation potential exists in the energy sector, which accounts for 63% of total potential under the Cost-effective Mitigation scenario (Fig. 4a) and 51% under the Deep Mitigation scenario (Fig. 4b). Agriculture offers a larger reduction potential under the Deep Mitigation scenario (41% of the total) than the Cost-Effective scenario (31%), while wastewater offers a 3% total reduction under the Deep Mitigation scenario. In the near term, and taking into consideration the shorter lifetime of methane emissions by using a 20-year GWP, potential methane reductions from the energy and agriculture sectors by 2030 are shown in Fig. 4c, d.

Based on a review of several studies of non-$CO_2$ mitigation measures—i.e., EPA[23], Höglund-Isaksson[29], Höglund-Isaksson et al.[30], and Yang et al.[31]—we estimated average marginal abatement costs for each methane mitigation measure. Applicable measures for China were screened based on our scenario definitions for abatement costs, and technical methane reduction potentials were then applied to applicable subsectors in our bottom-up model. Total reduction potentials for individual measures

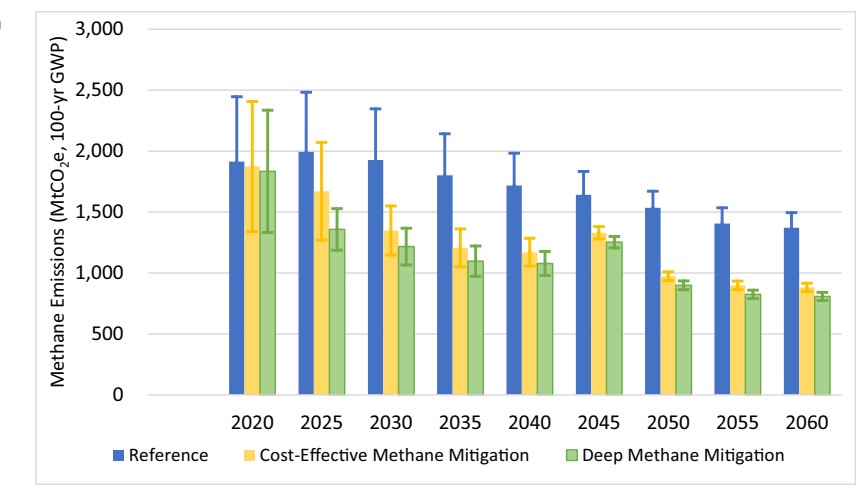

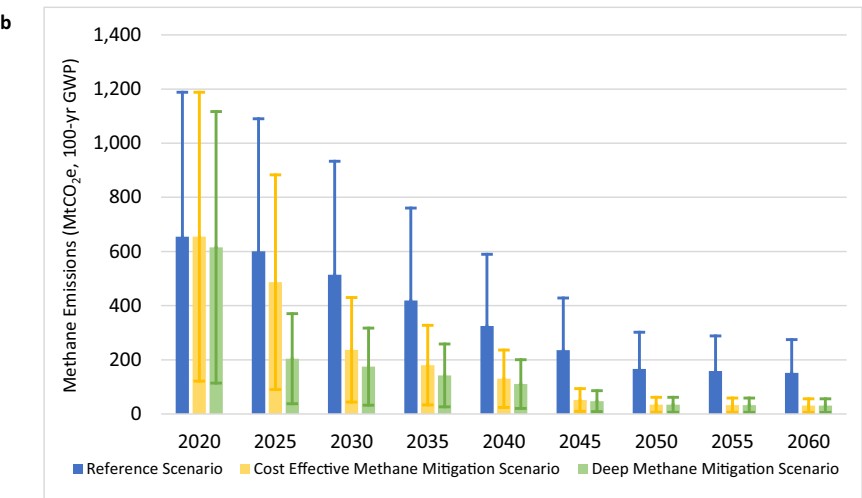

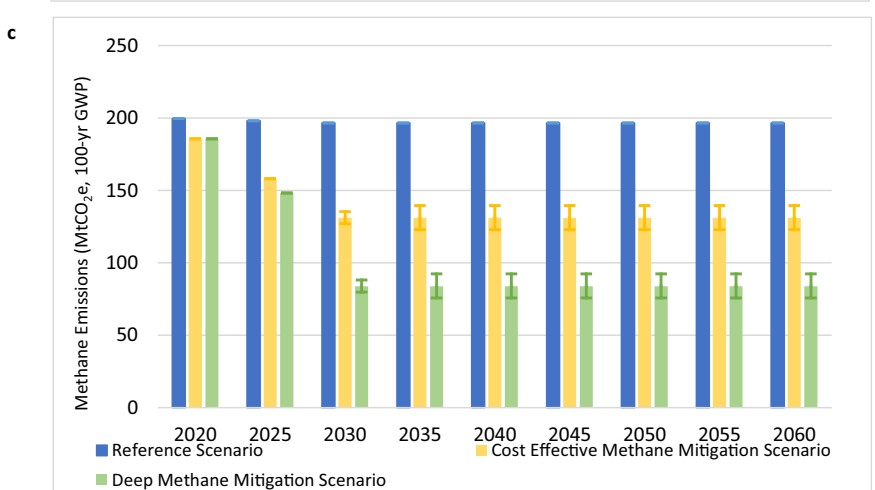

**Fig. 2 | China's methane emissions and uncertainties, 2020–2060. a** Total emissions and uncertainties; **b** coal mine emissions and uncertainties; and **c** rice cultivation emissions and uncertainties, uncertainties not shown for Reference scenario as only biochar mitigation efficacy is assessed and included only in mitigation scenarios. Error bars shown represent the high and low ranges of uncertainty assessed for selected sources. Source data are provided as a Source Data file.

were calculated using our model under both Cost-effective and Deep Mitigation scenarios. By combining average cost data (expressed in 2020 US $) for applicable measures from the literature with the methane reduction potential of specific measures calculated by our model, we derived a 2030 cost curve for the Deep Mitigation scenario Fig. 5) with sector-averaged costs. For all sectors (excluding manure management and biomass combustion), we find a total methane

reduction potential of 660 $MtCO_2e$ in 2030 with an average cost of −$6.40/$tCO_2e$.

Of all mitigation measures, switching from extensive and semi-intensive to intensive aquaculture systems has a negative average abatement cost—that is, a net benefit of $107/$tCO_2e$—to accompany its sizeable reduction potential of 104 $MtCO_2e$ in 2030. The higher stocking density of intensive aquaculture systems results in

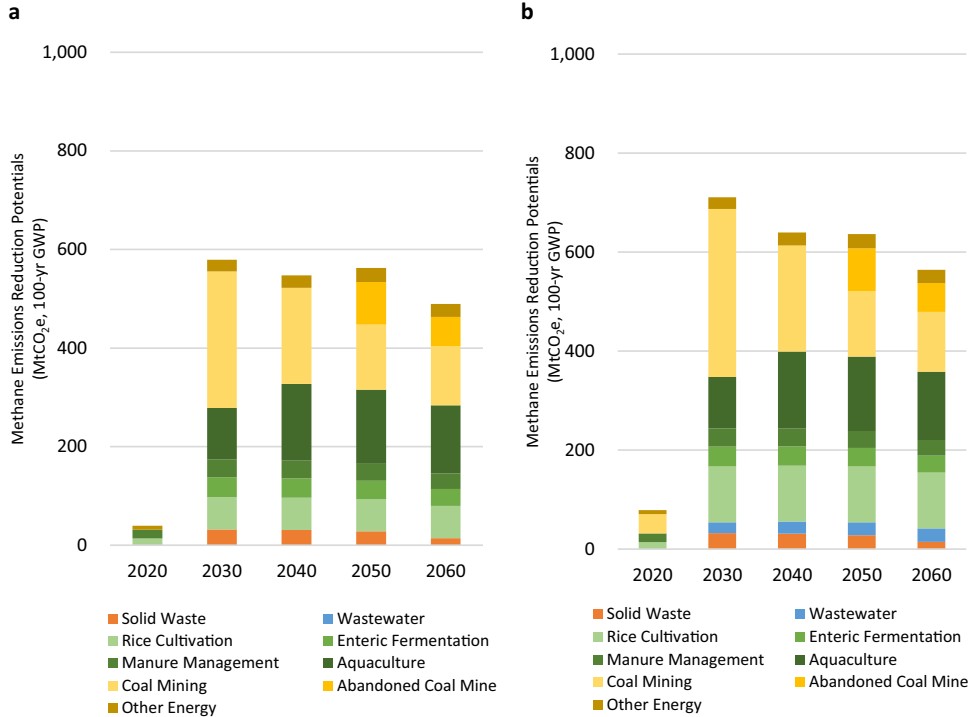

**Fig. 3 | Methane emission reductions by source. a** Cost-effective Mitigation scenario and **b** Deep Mitigation scenario. The methane emission reduction potential shown for both scenarios is relative to the Reference scenario without any mitigation measures. Source data are provided as a Source Data file.

significantly higher fish production and revenue that far offsets the additional costs of the system[32].

Biochar application and drainage in the agriculture sector, and utilizing higher-concentration coal mine methane for power generation in the energy sector, also have negative average abatement costs but a lower reduction potential compared to aquaculture. VAM in the coal mining sector has the largest reduction potential (223 MtCO$_2$e) and a relatively low average abatement cost of only $5/tCO$_2$e. Other mitigation measures with higher average abatement costs (i.e., above $25/tCO$_2$e) include water and fertilizer management in rice cultivation, onsite use of lower-concentration diluted coal mine methane that would otherwise be vented, oil sector mitigation, and enteric fermentation. Collectively, these higher cost measures provide another 164 MtCO$_2$e of methane reduction in 2030.

In 2060, the largest source of methane mitigation potential lies in the agriculture sector, which accounts for more than half of the total under both Cost-effective and Deep Mitigation scenarios (Fig. 6a, b). The energy sector—which accounts for 42% and 36% of total potential 2060 methane reductions under the Cost-effective and Deep Mitigation scenarios, respectively—represents the second-largest source of potential reductions, mainly through reductions from abandoned coal mines and coal production. By comparison, the waste sector contributes the smallest source of methane mitigation potential.

Figure 6c compares the remaining 2060 methane emissions by a source under all three scenarios against the base year of 2015. Under all scenarios, there is a notable decrease in both absolute and relative methane emissions from the energy sector, primarily owing to decreased emissions from coal mining. At the same time, there is an increase in wastewater methane emissions due to growth in drivers such as industrial activity along with a relative paucity of mitigation measures for that sector having costs below US$100/tCO$_2$e. Agricultural emissions in 2060 continue to account for half of total methane emissions, with most coming from enteric fermentation where limited mitigation measures exist. However, the total 2060 methane emissions from agriculture decrease with the deployment of

Deep Mitigation measures. These results emphasize the need for continued focus on methane mitigation opportunities in the agriculture and wastewater sectors, which represent the majority of remaining methane emissions in 2060—even when individual mitigation measures costing up to US$100/tCO$_2$e are fully deployed.

## Discussion

This paper uses a bottom-up modeling approach with updated activity driver projections and abatement cost analysis to account for additional methane emission sources, areas of uncertainties, and mitigation measures in the energy and agricultural sectors in China. Our modeling results underscore the significant cost-effective potential for reducing methane emissions in China by 2030, with 660 MtCO$_2$e reductions possible with average negative abatement costs of US$6.40/tCO$_2$e and 710 MtCO$_2$e possible if more uncertain and costly manure management and biomass measures are included.

Most mitigation potential in 2030 lies in the energy sector, primarily coal mining, but by 2060 this will shift to the agriculture sector following China's clean energy transition to meet its carbon neutrality goal. Despite coal's declining role in China's future energy sector, coal mining remains a key source of methane emissions and there are high uncertainties around coal mine methane emissions in the near term due to variations in emission factors linked to geographic and mine-specific conditions. Abandoned coal mine methane is another key source of emissions that has been considered in only a limited number of China studies, and possible mitigation measures are not yet well understood. To achieve the relatively low-cost mitigation potential identified for coal mining, the development of regulations and standards such as the high-concentration coal mine methane standards currently under revision can be considered. In addition, providing financial incentives through certified carbon credits under China's voluntary carbon market for China Certified Emission Reduction credits or other green financial measures can help accelerate the adoption of methane mitigation measures with moderate costs such as VAM mitigation. The emissions reduction impact of these certified

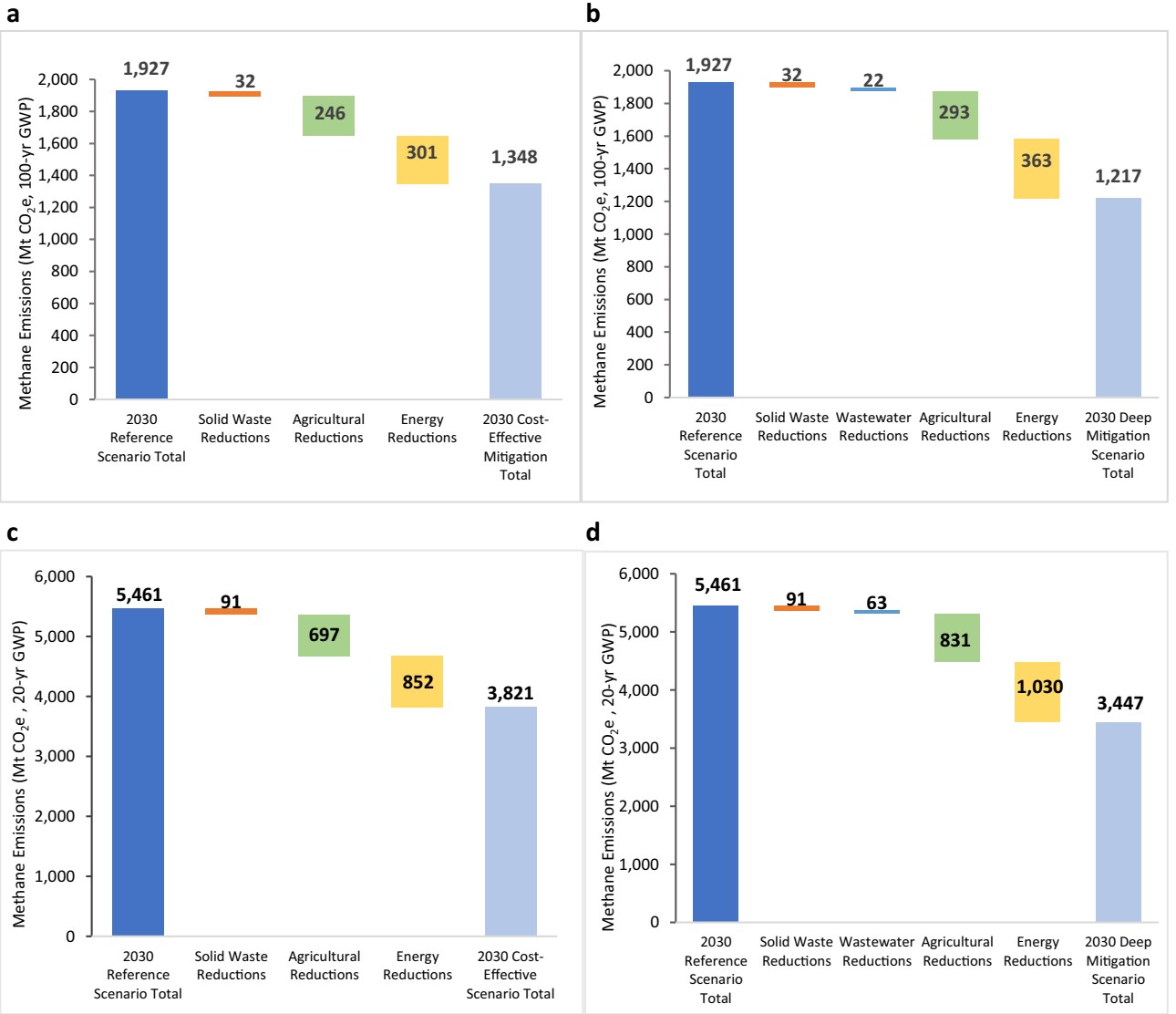

**Fig. 4 | 2030 Methane emissions reductions by sector and scenario.** Total methane emissions and emissions reduction chart shown for **a** Cost-effective Mitigation scenario; **b** Deep Mitigation scenario, with emissions expressed as MtCO₂e in 100-year GWP; **c** Cost-effective Mitigation scenario; and **d** Deep Mitigation scenario, with emissions expressed as MtCO₂e in 20-year GWP.

carbon credits can be strengthened by offset protocols developed to provide a standardized approach for reducing the risk of over-crediting emissions reductions from mine methane capture, such as the one developed in California in 2014[33].

In the agriculture sector, aquaculture and biochar applications in rice cultivation could reduce methane emissions at a negative cost (i.e., net benefit) and should be pursued more rigorously in both research and development and implementation to enhance their mitigation efficacies. China has historically had a strong agricultural extension program and this robust outreach model can be utilized to help raise awareness of the net economic benefits of adopting mitigation measures such as biochar and aquaculture. In the longer term out to 2060, the agriculture sector holds the greatest reduction potential but is also the largest source of remaining methane emissions in China. Notable remaining sources of agricultural methane emissions include enteric fermentation and rice cultivation, for which additional and novel mitigation measures may need to be assessed. Long-term government financing, as well as potentially increased private investments through the inclusion of agricultural projects in China's re-started voluntary carbon market, can provide the research and development funding

needed to increase commercial viability and deployment. Additional mitigation measures must also be explored for the wastewater sector, a growing source of methane emissions where current measures are limited and costly. For these hard-to-abate methane sources, international collaboration on research and development can help address remaining methane emission sources and high-cost mitigation opportunities.

This paper addresses some of the largest uncertainties around projected methane emissions from coal mining and rice cultivation in China, and highlights lesser-known sources and mitigation opportunities in the energy and agriculture sectors. However, we did not consider or attempt to quantify uncertainties for other subsectors, such as waste and wastewater, whose methane emissions are expected to grow, or for enteric fermentation. We also focused our mitigation analysis on existing and commercialized mitigation measures with abatement cost data and did not model emerging technologies for methane mitigation or broader socio-economic paradigm shifts such as dietary or other lifestyle changes. Lastly, our projections of key activity drivers could be further improved with sensitivity analysis. These topics will be included in our future research.

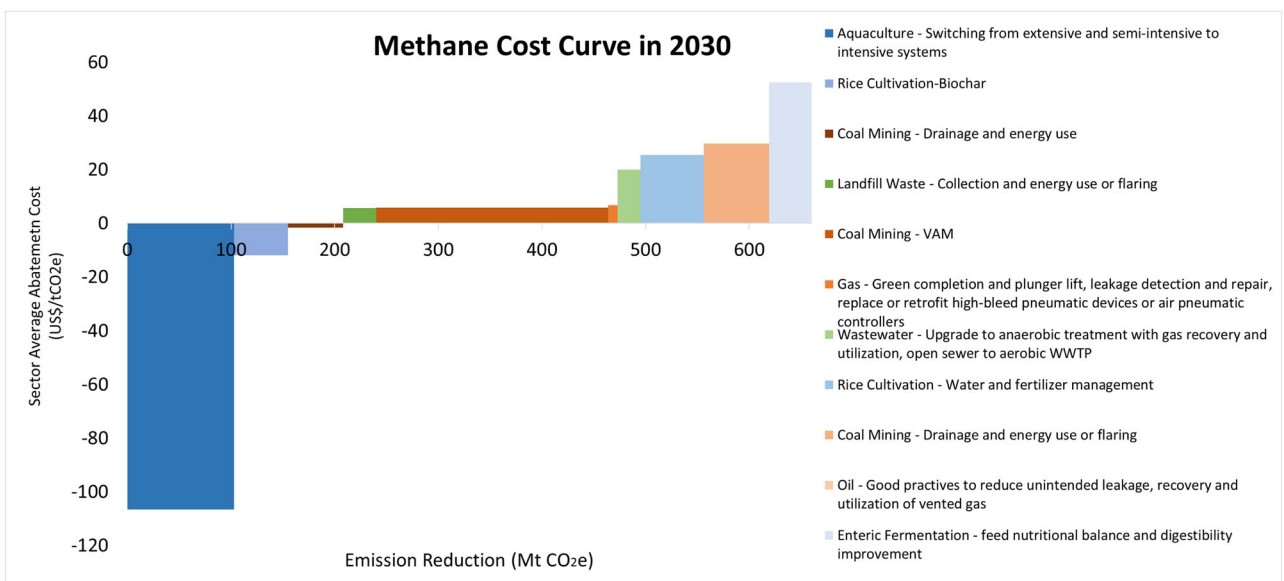

**Fig. 5 | 2030 Methane mitigation cost curve for deep mitigation scenario.**
Reductions in methane emissions from manure management and biomass combustion are not shown due to high and uncertain costs. Abatement costs shown for each subsector are the average costs over a mix of mitigation measures with different individual costs, expressed in 2020 U.S. dollars ($) with an equivalent conversion rate of 1 US$ = 6.90 Chinese yuan. VAM is ventilation air methane, WWTP is wastewater treatment plant. Source data are provided as a Source Data file.

## Methods

### Modeling framework and approach

This study uses a bottom-up modeling approach to analyze in detail the anthropogenic methane emission sources in energy and non-energy sectors from a techno-economic perspective and to evaluate future pathways to lower methane emissions. In a bottom-up modeling approach based on the IPCC Guidelines for estimating production-based greenhouse gas emissions, specific methane-emitting actions and mitigation technologies are modeled at the level of specific emission sources, such as coal mining or enteric fermentation[34]. Methane emissions are calculated as a product of the emitting activity source and a source-specific emission factor that could change with the market adoption of mitigation measures or technology. The modeling methodology described below is directly adapted from the methodology used by the IPCC[34] and the U.S. EPA in their respective Guidelines for Greenhouse Gas Emission Inventories[23]. More specifically, following the earlier bottom-up modeling methodology developed in ref. 24, baseline methane emissions without any methane mitigation measures were calculated from source $s$ in year $t$ as the activity data $A_{st}$ times an unabated emission factor $EF_{st}$ for the reference scenario using Eq. (1):

$$E_{st} = \sum [A_{st} \cdot EF_{st}] \tag{1}$$

If emissions are controlled through the implementation of a mitigation technology or measure $m$ under the cost-effective mitigation and deep mitigation scenarios as defined below in Eqs. (2) and (3), the fraction of the emissions controlled, $E_{stm}$, is specified by $Red_{stm}$.

$$E_{stm} = \sum [E_{st} \cdot Red_{stm}] \tag{2}$$

Where

$$Red_{stm} = \sum_{m=0}^{n} [MP_m \cdot TRE_m] \tag{3}$$

And where $A_{st}$ is the emission source activity (e.g., number of people, number of animals, amount of fuel, etc.), $EF_{st}$ is the unabated emission factor per unit of activity for that specific activity source, $Red_{stm}$ is the reduction efficiency for that activity source $s$ in year $t$ for measure $m$, where reduction efficiency is calculated as the market adoption rate of a specific methane mitigation measure $m$, $MP_m$, times that methane mitigation measure's technical effectiveness for reducing emissions, $TRE_m$. In some sectors where mitigation measures can be combined with additive reduction efficiencies, $Red_{stm}$ is the sum of all applicable measures.

Following this bottom-up modeling methodology and earlier data collected in ref. 24, a bottom-up spreadsheet model is used to account for all energy and non-energy sources of methane emissions in China by tracking activity drivers and unabated methane emission factors for each source and projecting changes over time under defined scenarios (see Table 1). Figure 7 illustrates a simplified diagram of the modeling approach.

### Data sources and assumptions for key modeling inputs

For historical data, the bottom-up spreadsheet model inputs were calibrated with the latest reported national statistics for both energy and non-energy activity variables such as population, households, industrial production, agricultural activity, and waste generation. Energy consumption data by fuel and by sector also were calibrated to the latest published national energy balances. Most of the historical activity data series were derived from official national statistics, which have been critiqued for deficiencies in methodology and transparency with examples of data inconsistency and discrepancies, particularly for economic indicators[35,36]. However, in the absence of alternative data that can be verified, we followed the official statistics as they are also the basis for China's UNFCCC inventory reporting and our estimates for 2020 methane emissions are in line with other independent estimates as seen in Fig. 1. Calculations of energy-related emissions data used China-specific fuel energy content and for non-energy methane emissions, a mix of emissions factors from the 2010 Chinese Provincial Guidelines for GHG Inventories, other sector-specific Chinese studies for methane[31], published literature with updated, sector-specific emission factors for China[13,15,23] and the 2006 IPCC Guidelines for GHG Inventories.

The calculated historical emissions were compared against methane emissions reported in the two latest years—2014 and 2017—of

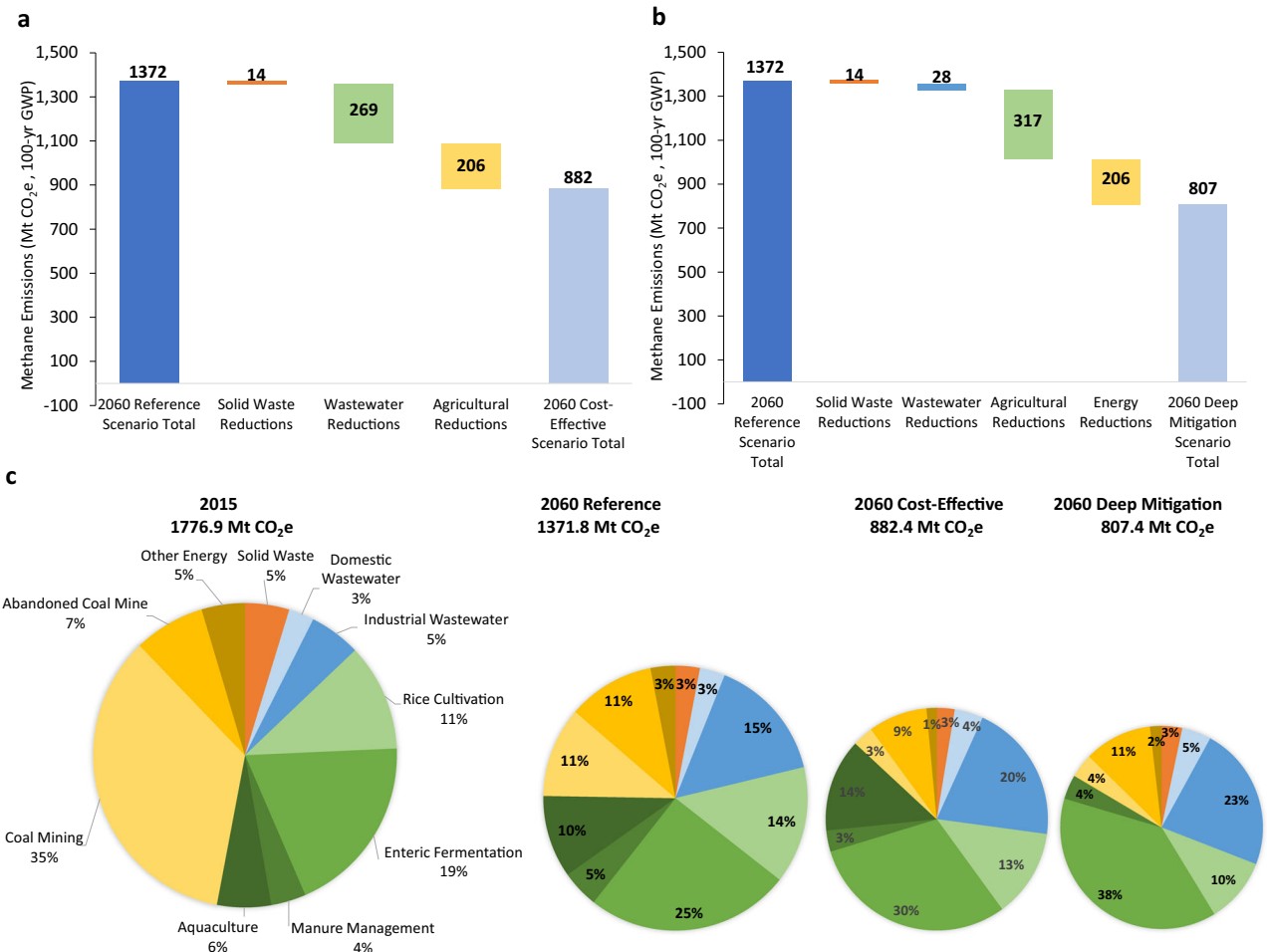

**Fig. 6 | 2060 Methane emissions reductions by sector and scenario.** 2060 methane emissions reduction potential shown by sector in MtCO₂e (100-year GWP) relative to Reference scenario for **a** Cost-effective Mitigation scenario; **b** Deep Mitigation scenario; and **c** comparison of 2015 and 2060 remaining methane emissions by subsector and scenario, with the size of the pie chart scaled to the magnitude of total methane emissions.

China's national GHG emissions inventory submitted in its 2019 and 2023 National Communication to the UNFCCC (see Supplementary Table 1 for values) to ensure close calibration.

For both energy and non-energy sectors, future activity projections were also extended through 2060 based on the latest studies, input from sectoral experts on activity drivers, and assumptions about future growth trends as summarized in Table 1 and detailed in Supplementary Table 2. Energy-related activity projections for the coal, oil and natural gas, power, and energy-consuming sectors of buildings, industry, and transport were taken from the latest scenario analysis conducted by another bottom-up, China energy demand model prior to 2050, as well as other China-based energy models from 2050 to 2060[25,37–39]. For abandoned coal mines, the global rate of abandonment used by Kholod et al.[40] and subsequent China-focused studies[13,15] is also used due to persisting data gaps in China-specific abandoned mine depths, timing of abandonment, and decay curves. Non-energy methane emissions are calculated in the model for the major emitting sectors of agriculture and waste and wastewater. Non-energy-consuming activities in these sectors, such as rice cultivation, livestock management, wastewater, and solid waste generated are expected to increase in many cases as China continues its economic transition. While historical trends and insights from sectoral experts are incorporated into the future activity projections, the projections for non-energy activity drivers notably do not include significant paradigm shifts that may be needed for deep decarbonization due to limitations from uncertainty on timing and scalability of these shifts.

For example, dietary changes, food waste reduction, and other lifestyle changes could significantly affect future activity drivers such as growth trends in livestock population, rice and aquaculture demand, and waste diversion trends, but these are not considered in our current model.

Methane emission factor data are taken from a variety of sources and with additional analysis. As detailed in Table 1, the vast majority of emission factors used are China-specific and based on the latest published literature where possible. In the few instances where China-specific emission factors were not readily accessible (e.g., primarily for oil and gas production, transmission and distribution, and combustion and solid waste), IPCC default data were used.

For emission factors, additional analysis was conducted to account for data gaps and uncertainties in methane emissions from coal mining and rice cultivation. For coal mine methane, the upper and lower bounds for the emission factor were derived by considering uncertainties around province-specific median emission factors reported in Sheng et al.[13]. The calculation assumes constant future shares between high and low methane content production in the absence of additional information. An implied uncertainty range of ±80% was identified based on uncertainty ranges around the median emission factor for the three largest-producing provinces of Inner Mongolia, Shanxi, and Shaanxi, which account for over 60% of current coal production (based on province-specific data from refs. 13,14).

For rice cultivation, the baseline emission factors are determined according to the average emission factors for each type of growth

**Table 1 | Activity drivers, activity projections, and emission factors for China's methane emission sources**

| | Activity drivers | Basis for activity projections | Unabated emission factor | Basis for emission factors |
|---|---|---|---|---|
| **Energy sector** | | | | |
| Coal mining | Coal mining production, assuming current underground and surface mining shares remain constant | Endogenously calculated in China energy models based on demand projection and resource availability, and varies by scenario based on pace of energy transition modeled[25,37,39]. | 0.27 g CH$_4$/MJ | The author calculated median EF based on province-weighted production and province-specific median EF for the three largest provinces from Sheng et al.[13] |
| Abandoned coal mine methane | Abandoned coal mine capacity | The 4.7% annual abandonment rate of annual coal production capacity and year-on-year decline in coal production capacity (if any) is assumed to be abandoned, based on the methodology described in ref. 40 and previously applied to China in refs. 13,15. Varies by scenario based on the pace of transition away from coal modeled. | Varies by year, ranges from 0.026 g to 0.095 g CH$_4$/g coal | Implied annual EF calculated using data from Chen et al.[15] Method 2 calculation by dividing total abandoned methane emissions for Chinese coal mines by total mean abandoned mine capacity. |
| Oil and gas production | Oil and natural gas production output | Endogenously calculated in China energy models based on demand projection and resource availability[25,37,39]. Oil and gas production do not vary by scenario because China is predominantly a net oil importer with limited domestic production in later years due to resource availability. | Crude oil: 0.871 g CH$_4$/GJ produced Natural gas: 0.32 g CH$_4$/MJ produced | Tier 1 default EFs from IPCC.[34] |
| Oil and gas transmission and distribution | Oil and natural gas demand for T&D | Endogenously calculated in China energy models based on demand projection and resource availability and varies by scenario based on pace of energy transition modeled[25,37,39]. | Crude oil: 0.148 g CH$_4$/GJ Natural as: 0.074 g CH$_4$/MJ | Tier 1 default EFs from IPCC[34] using average values for extraction transmission and distribution. |
| Biomass combustion | Rural biomass combustion for heating and cooking | Biomass demand is calculated endogenously by a model based on fuel shares, heating, and cooking energy demand[37]. Varies by scenario based on the pace of energy transition modeled. | 229 kg CH$_4$/TJ biomass combusted | China-specific EF, weighted by firewood and stalk shares from China's 2010 GHG Inventory for Provinces Guidelines.[41] |
| Transport diesel and gasoline vehicles | Diesel and gasoline consumption for different vehicle types and weight classes: trucks, buses, passenger vehicles | Fuel consumption by vehicle type is calculated endogenously by the China demand model based on fuel shares, efficiency, and mobility demand[37]. Varies by scenario based on the pace of energy transition modeled. | Heavy-duty gasoline vehicles: 25 kg CH$_4$/TJ Light-duty gasoline vehicles: 3.8 kg CH$_4$/TJ Diesel vehicles: 3.9 kg CH4/TJ | Tier 1 default average values from IPCC.[34] |
| **Agriculture sector** | | | | |
| Rice cultivation | Harvest area for four different growth seasons of single season growth, double season harvest spring, double season harvest fall, and flooded winter non-growing seasons | Projected growth rates based on China Agriculture Outlook's[47] growth through 2030 and then assumed constant after 2030. | Varies by growth season, ranging from 196 kg to 300 kg CH$_4$/hectare | Median EFs for four main types of growth season from China's 2010 GHG Inventory for Provinces Guidelines.[41] |
| Enteric fermentation | Total livestock of cattle, buffalo, horses, donkeys, mules, camels, hogs, sheep, and goats | Varies by animal type: growth based on China Agriculture Outlook's[47] growth rates through 2024, then driven by population through 2060 assuming constant livestock per capita demand. | Varies by animal type, ranging from 1 kg to 76 kg CH$_4$/head | Median EFs for each type of livestock from China's GHG Inventory for Provinces Guidelines.[41] |
| Manure management | Total livestock of cattle, buffalo, horses, donkeys, mules, camels, hogs, sheep, and goats | Varies by animal type: growth based on China Agriculture Outlook's[47] growth rates through 2024, then driven by population through 2060 assuming constant livestock per capita demand. | Varies by animal type, ranging from 0.34 kg to 4.7 kg CH$_4$/head | Median EFs for each type of livestock from China's GHG Inventory for Provinces Guidelines.[41] |
| Aquaculture | Freshwater aquaculture production | Aquaculture demand projection driven by population and per capita aquaculture demand (assumed to stay constant after 2030) | 0.116 Tonne CH$_4$/tonne aquaculture production | Calculated EF based on 2014-15 emissions and production data from Yuan et al.[48] |
| **Waste and wastewater sector** | | | | |
| Solid waste | Solid waste disposed | Based on regression of solid waste generated as a function of per capita GDP, with population as growth driver. | Emissions calculated using China's solid waste disposed activity projection (assuming no change in waste composition) as input to IPCC First Order Decay Model.[34] | Emissions calculated using the IPCC First Order Decay Model from IPCC.[34] |

**Table 1 (continued) | Activity drivers, activity projections, and emission factors for China's methane emission sources**

| | Activity drivers | Basis for activity projections | Unabated emission factor | Basis for emission factors |
|---|---|---|---|---|
| Domestic wastewater | Organics in wastewater as a function of population and urbanization | Calculated as population times per capita biochemical oxygen demand (BOD) using Asian regional defaults from IPCC.[34] | 0.08 kg $CH_4$/kg BOD | China-specific EF taken from Yang et al.[31] |
| Industrial wastewater | Chemical oxygen demand (COD) in wastewater | Total COD produced calculated based on regression analysis of historical COD produced to total industrial value-add GDP. | 0.12 kg $CH_4$/kg COD | China-specific EF taken from Yang et al.[31] |

season reported in China's 2010 Provincial Greenhouse Gas Inventory Guidelines[41]. These baseline emission factors may have incorporated some effects of the existing straw returning to the field. However, due to the lack of data, the straw return ratio in the base year and its impact on baseline emission factors are hard to quantify.

For cost-related data, we conducted a literature review on cost analysis of methane mitigation measures and identified several studies that have estimated those costs specifically for China, including two reports from the U.S. Environmental Protection Agency[23,42], two studies from the International Institute for Applied Systems Analysis[29,30] and one analysis from China Environmental Press[31]. Information from the International Energy Agency's methane tracker is also included[43]. Based on our specific scenario definitions, we screened applicable mitigation measures for China following the marginal abatement costs defined in the two scenarios and applied the identified market penetration rate and technical reduction effectiveness to applicable sectors to calculate the emission reduction efficiency used in our bottom-up model. The assumptions and data used are detailed in Supplementary Table 3.

### Future methane emission scenarios

We developed three main scenarios to assess the possible growth in China's methane emissions and mitigation potential through the adoption of different technologies, measures, and strategies under two plausible trajectories defined primarily by the marginal abatement costs of individual mitigation measures. In the absence of specific quantitative policy targets for aggregate or sectoral methane reduction, we used marginal abatement cost estimates as the basis to evaluate the potential application and scale of deployment and effectiveness of mitigation measures for each methane source sector from 2021 through 2060.

The scenarios evaluated include a Reference scenario of a clean energy transition and decarbonization consistent with Zhou et al.[37]'s current policy energy transition scenario to serve as a baseline scenario that does not consider any methane mitigation measures in the absence of any clear quantitative targets or clearly defined actionable policy actions for methane emissions reduction in methane-related policies to date. We also used a Cost-effective Mitigation scenario that assumes the same pace of energy transition as the reference scenario but with full adoption of individual cost-effective methane mitigation measures with current abatement costs below $10/$tCO_2$e by 2050. This Cost-effective Mitigation scenario reflects a more plausible scenario with low to moderate marginal abatement costs of under US$10/$tCO_2$e, a level that is similar to the recent average carbon price for China's emissions trading scheme for carbon. Lastly, we used a Deep Mitigation scenario that assumes an accelerated clean energy transition and faster and earlier adoption of cost-effective methane mitigation measures by 2025, and the application of additional, higher-cost individual mitigation measures with current abatement costs below US$100/$tCO_2$e. This scenario represents a technically feasible scenario that would require significantly greater policy actions and investments in order to accelerate the clean energy transition and adoption of low to moderate-cost methane mitigation measures, and spur the adoption of higher-cost methane mitigation measures at a cost threshold consistent with that used in other global analysis (e.g., EPA[23]).

We conducted a literature review of China-specific cost analysis of methane mitigation measures and screened applicable mitigation measures based on the marginal abatement costs (<US$10/$tCO_2$e and <US$100/$tCO_2$e) defined in the two scenarios and applied the identified technical reduction potential to applicable sectors in our bottom-up model. Table 2 summarizes key differences between the two mitigation scenarios in terms of measures deployed and pace of adoption based on the cost analysis. Detailed descriptions of the mitigation measures considered, assumptions, and estimated costs of abatement for both mitigation scenarios are outlined in Supplementary Table 3.

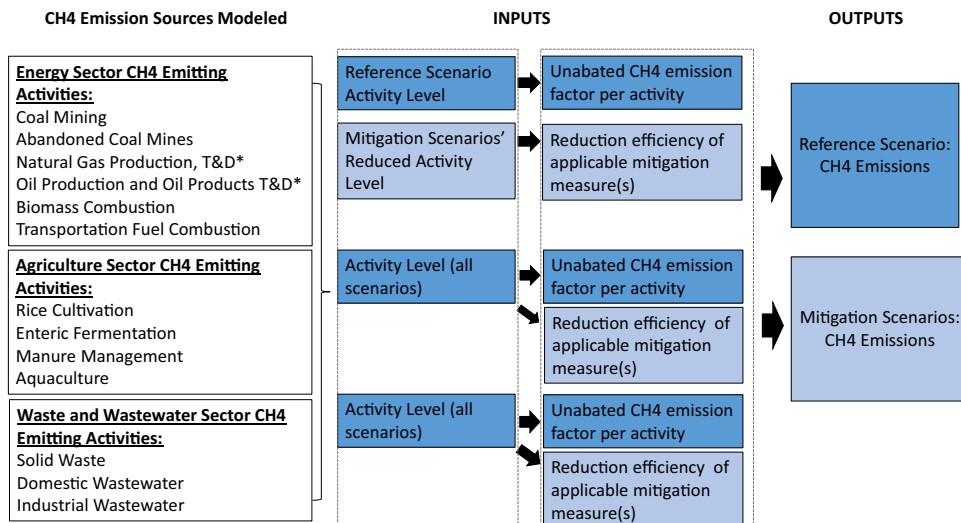

**Fig. 7 | Simplified bottom-up methane modeling framework structure.** *T&D is transmission and distribution. Applicable mitigation measures are selected based on scenario cost thresholds and the China-specific marginal abatement cost of individual mitigation measures. The dark blue shade represents inputs to the Reference scenario and the light blue shade represents inputs to the two mitigation scenarios.

**Table 2 | Summary of methane mitigation measures by mitigation scenario**

| | Measures: Cost-effective Mitigation scenario | Additional measures: Deep Mitigation scenario |
|---|---|---|
| **Energy sector** | | |
| Coal mining | Ventilation air methane (VAM) oxidation for high/medium/low concentration gas; gas collection and flaring; gas collection for energy use. | Accelerated full adoption of measures by 2025. |
| Abandoned coal mine methane | Indirect mitigation only: gradual phasedown of coal and subsequently abandoned mines over time due to clean energy transition. Direct mitigation measures are not considered due to a lack of data on applicability, costs, and technological immaturity. | Indirect only: faster phasedown of coal and subsequently abandoned mines due to faster clean energy transition. Direct mitigation measures are not considered due to a lack of data on applicability, costs, and technological immaturity. |
| Oil production, transmission, and distribution | None, due to higher costs. | Practices to reduce unintended leakage; recovery and utilization of vented gas. |
| Natural gas production, transmission and distribution | Green completion and plunger lift; leakage detection and repair; replace or retrofit high-bleed pneumatic devices or air pneumatic controllers. | Accelerated full adoption of measures by 2025. |
| Biomass combustion | Biomass consumption is reduced due to the clean energy transition. | Greater reduction in biomass consumption due to greater building electrification. |
| Transport diesel and gasoline vehicles | Diesel and gasoline consumption are reduced due to transport electrification. | Greater reductions in diesel and gasoline consumption due to full transport electrification. |
| **Agriculture sector** | | |
| Rice cultivation | Cost-effective irrigation practices; biochar application. | Greater and earlier deployment of changed irrigation practices. |
| Enteric fermentation | Improving the nutritional balance of livestock feed and feed digestibility. | None |
| Manure management | Converting manure to compost. | None |
| Aquaculture | Switching from extensive and semi-intensive to intensive systems. | None |
| **Waste and wastewater sector** | | |
| Solid waste | Gas collection and flaring; gas collection for power generation. | None |
| Domestic wastewater | None, due to higher costs. | Upgrade to anaerobic treatment with gas recovery and utilization; open sewer to aerobic wastewater treatment plant (WWTP). |
| Industrial wastewater | None, due to higher costs. | Upgrade to anaerobic treatment with gas recovery and utilization; open sewer to aerobic wastewater treatment plant (WWTP). |

Biochar is an important mitigation strategy with notable potential to reduce methane emissions from rice cultivation, with crop residues potentially serving nearly half of biochar feedstocks at a national level[19]. However, the role of biochar in overall methane mitigation for China has received limited attention in existing multi-sector methane mitigation analyses. The current biochar utilization rate is low, but due to data availability, we made the simplifying assumption that biochar utilization is zero until 2020. There is some uncertainty surrounding biochar's methane reduction efficiency. According to the meta-analysis of 113 experimental observations in ref. 20, biochar's average methane reduction efficiency is 26.35%, with a 95% confidence interval ranging from 22.01% to 30.47%; our study adopts these findings when estimating biochar's methane mitigation potential. Our study also estimates the per unit methane reduction cost of biochar,

taking into consideration uncertainties in biochar application rate, per unit cost, methane reduction efficiency, and yield improvement potential. For the biochar application rate, we assume a rate of 2.8 tonnes per hectare, following Nan et al.[44]. For the cost of biochar application, we use the average of cost estimates for biochar from crop straw reported by Mohammadi et al.[45][45] and Clare et al.[46], or $115 per tonne. Finally, for yield improvement potential, we use the range reported by Xia et al.[19], which incorporates 230 experimental observations from various studies. The average yield improvement potential is 11%, and the 95% confidence interval ranges from 9% to 13%. Considering uncertainties in the biochar application rate, per unit cost, methane reduction efficiency, and yield improvement potential, we estimate the per unit methane mitigation cost of biochar to be −$14/tCO$_2$e on average, with a range from −$47/tCO$_2$e to $29/tCO$_2$e.

## Data availability

The data generated in this study are provided in the Supplementary Information. Source data are provided with this paper.

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

## Acknowledgements
The authors acknowledge funding support from the Global Methane Hub under grant 018215-2022-08-01 (J.L.) for this work.

## Author contributions
N.K. and J.L. conceptualized the analysis. N.K. wrote the paper, and together with J.L., X.L., and W.W. carried out data collection and analysis. N.K., J.L., X.L., and W.W. contributed to the discussion and revision of the paper.

## Competing interests
The authors declare no competing interests.
