## [Peer Review File · Nature Communications]

Reviewers' comments:

Reviewer #1 (Remarks to the Author):

Methane is the second most important anthropogenic greenhouse gas after carbon dioxide. The paper aims to China's methane mitigation potential through 2060. It addresses an important and timely topic in the field. This work is meaningful. However, the paper does not carry the punch that it should.

Here are my specific comments:

In general, the persuasiveness of the results is not sufficient due to the inadequate and unreliable original data, especially the mitigation cost data. The emission sources of China's CH₄ are very complex. Incorrect information in the source attribution have been always used in previous studies. If this study has added and revised the estimations of different emissions sources, the authors should point out incorrect methods, emission factors and activity-level data used in such studies in detail.

The methodology and data sources haven't been clearly described.

Some details of emission data are not provided as the basic information for the mitigation scenarios. Are details available on whether acceptance data were measured (e.g., with a scale) or estimated?

There needs to be enough information in this paper, either in the main text or supplementary information.

More details about the methodology/analysis would help evaluate appropriateness and reproducibility. The scenarios were defined by the costs prior to 2050 rather than 2060, which needs explanations.

☒ L150, ...destroys or utilizes ventilated air methane (VAM) released from coal mines by 2050, why is 2050? Moreover, this claim is made without sufficient justification.

☒ L190,...utilizing higher-concentration coal mine methane for power generation in the energy sector, also have negative average abatement costs but a lower reduction potential. It doesn't seem to make sense and needs more explanations.

The inventory developments of different sources and the impacts and effects of cutting emissions during the social transition should be addressed in depth. It is more challenging to address the stochastic and uncertain variables and constraints of the problem than addressing the deterministic ones. It is not clear what each of the mitigation options consist of. There should be more description of each scenario and quantitative information about each scenario, with citation to the source of each quantitative variable or dimension.

For instance, in Additional Measures: Deep Mitigation Scenario, the methane mitigation measures of abandoned coal mines are earlier phasedown due to faster clean energy transition, which is

unclear statement, needs more explanations. How were the emission mitigation measures for methane emissions from abandoned mines considered in the scenario setting of the paper? The paper used an abandonment rate of 4.7% for abandoned coal mines in China, how applicable is this parameter to abandoned coal mines in China?

The discussion could be strengthened by further interpreting results in relation to previous research, especially the mitigation potential in specific sector. The author didn't give a clear explanation and critical discussion about the policy implications of such study. More intuitively expressions like how to deal with China's CH₄ emissions in the future could be discussed.

Limitations of the study design should be more critically addressed, such as the methodology of projections of key activity driver.

The figures are presented roughly, such as Figure 4.

Proofreading for typos would improve the clarity of writing, especially in L139-144 and 172. The manuscript will benefit from a thorough language check.

Reviewer #2 (Remarks to the Author):

Summary

The authors present estimates of China's methane emissions using a bottom-up process, where emissions are a function of economic activities (activity factors) and the estimated emissions intensity of the activities (emissions factors). The authors then present estimates of China's methane inventories in five-year time steps between 2020 and 2060 under two mitigation scenarios. The authors conclude there are significant opportunities for methane mitigation in China over the period of examination.

Comments

1. Research Question, Relevance and Overall Contribution

Open-access and peer-reviewed information on China's emissions inventory (including methane) is important, particularly as China does not produce a National Inventory Report for the UNFCCC. NIR methodology is also imperfect, as the UNFCCC methodological guidance is often behind the best-available science. For that reason, this paper potentially offers an important contribution to knowledge. In essence, this is an important avenue of academic research, as it improves understanding and measurement of an important climate-forcing GHG. However, I have major concerns regarding this paper's relevance and contribution in its current form, in part because of modelling choices and lack of clarity on assumptions.

A. The research question is currently unclear and not explicit. The authors do state what the paper does (“use a bottom-up modeling approach with updated methane emission projections and abatement cost analysis to account for additional sources, uncertainties, and mitigation measures in China’s energy and agricultural sectors”). However, without an explicit question to guide the reader (and why the methods are appropriate to answer it), the authors are not clearly identifying their purpose and contribution to knowledge.

B. I find it surprising that the authors do not comment on a lack of official Chinese emissions inventory (China does not submit a National Inventory Report to the UNFCCC) as part of the motivation for their work. I am unfamiliar with China’s emissions inventory process (if it exists), and it would be helpful for the authors to discuss this and contrast their inventory approach. This is particularly important given NIR methodology for methane is based on the IPCC fourth assessment report, while science has evolved significantly.

C. The authors could and should give a more fulsome discussion of China’s emissions policies. The discussion ignores China’s submissions to the UNFCCC, and couches Chinese policy as responses to US action. Given the goal of the paper is to inform both measurement and mitigation of methane in China, US policy is largely irrelevant whilst a discussion of Chinese policy is important for understanding the context and contribution of the article.

D. The authors motivate the paper by noting “increasing recognition” that limiting global temperature increases and climate change requires mitigation of GHGs other than carbon dioxide. I find this a false premise; my understanding is that policy focus on carbon dioxide is due to ease of measurement and availability of mitigation options in contrast to other GHGs. Moreover, recent scientific evidence on extensive under-measurement of methane emissions has underscored the imperative nature of addressing it, prompting policy action.

2. Assumptions, Methods, and Results

I have major concerns that make me question the paper’s relevance and contribution, related to the modelling and underlying assumptions, which I will discuss in detail below.

Without a clear research question and a discussion of China’s current inventory methods, alongside clear and transparent analytical methods (all are lacking in the paper as it stands), it is difficult for the reader to understand the contribution. A broad point, regarding exposition, is that the authors do not explicitly describe key features of the model and key assumptions. This is true for both the methane emissions inventory exercise and the mitigation analysis. Accordingly, there is inconsistency in explanation, and these choices makes it nearly impossible to interpret the overall trends and specific results. This means the paper is not particularly accessible, and not replicable.

A. The authors state their work “builds on previous work and provides new analysis by using a bottom-up modeling approach with updated assumptions about macroeconomic and physical emissions drivers.” The authors fail to describe the exact nature of these updated assumptions and how they differ from extant literature. Table 1 describes at a very high level the activity drivers, the activity projections, and the emissions factors. Figure 1 compares different estimates of China’s 2020 methane emissions. However, the authors do not report their actual activity factors or emissions factors or the methodology for determining both. Without this information, it is impossible to evaluate or verify the accuracy of their reported methane emissions. To address this, the authors should include a table with activity factors and emissions factors for each sector, with sources for each assumption. Ideally, the authors would also justify their assumptions relative to available academic literature and IPCC guidance.

B. The authors reference Lin et al. (2019) as the model underlying their results. However, the authors do not even summarise the model approach (beyond stating it is bottom-up). I find this wholly insufficient and strongly urge the authors to, at the very least, outline key equations and assumptions.

C. The authors spend a significant amount of page space on future methane trajectories under three scenarios (reference, cost-effective reductions, and deep mitigation). The underlying assumptions of these scenarios are unclear and undermine the contribution of the paper. Specifically, the reference scenario is “a clean energy transition and decarbonization consistent with Zhou et al. (2022).” This paper is missing from the references and so I was unable to review its assumptions. I do wonder why the authors didn’t just define a “current policy” reference case based on China’s existing policy documents, such as China’s NDC or the Mid-Century Long-Term Low Greenhouse Gas Emission Development Strategy (<https://unfccc.int/documents/307765>). Secondly, the authors define the cost effective mitigation scenario as mitigation measures “below [US] \$10/tCO_{2e} prior to 2050”; what is the rationale for this threshold? Is this real 2023 dollars or a nominal threshold? Similarly, what is the basis for defining deep mitigation as abatement costs below \$100 US/tCO_{2e}?

D. The authors also fail to source/cite the methane abatement cost estimates. While the authors do note their costs are “[b]ased on a review of several studies of non-CO₂ mitigation measures and abatement costs – i.e., EPA (2019), Höglund-Isaksson (2012), Höglund-Isaksson et al. (2020), and Yang et al. (2014),” this is insufficient detail. Moreover, the authors do not address whether these cost estimates are directly applicable to China. My concern is that a global cost estimate or a country-specific estimate for another country is not directly translatable to a cost estimate for China with different institutions and economic activity.

E. The authors also discuss uncertainties in methane emissions estimates, but their discussion lacks precision. Specifically, there can be uncertainty in both the activity factors and the emissions factors, and the authors give some examples but do not have a direct comparison of the uncertainty inherent in their approach vis a vis others. Moreover, this uncertainty can come from accuracy (closeness of estimates to the true value) or from precision (range of estimates likely resulting from repeated measurement); the authors do not disclose which source of uncertainty underlies their and others' modelling efforts. This is also relevant for the mitigation scenarios.

Minor comments

1. Given the paper's focus on methane, I do not understand why the authors present their results in tonnes of CO₂e.
2. In Figure 1, the authors could make more explicit their estimate (LBNL) relative to the others.
3. Why is the "other" missing in the LBNL estimate in Figure 1? What is included in "other"?
4. There are several cross-reference errors in the text.
5. The paper is poorly formatted, with fragments on text in some places (e.g., line 204 the paragraph starts with the fragment "Figure 6A").

Reviewers' comments:

Reviewer #1 (Remarks to the Author):

Methane is the second most important anthropogenic greenhouse gas after carbon dioxide. The paper aims to China's methane mitigation potential through 2060. It addresses an important and timely topic in the field. This work is meaningful. However, the paper does not carry the punch that it should.

Here are my specific comments:

In general, the persuasiveness of the results is not sufficient due to the inadequate and unreliable original data, especially the mitigation cost data. The emission sources of China's CH₄ are very complex. Incorrect information in the source attribution have been always used in previous studies. If this study has added and revised the estimations of different emissions sources, the authors should point out incorrect methods, emission factors and activity-level data used in such studies in detail.

Thank you for this comment. We respectfully disagree with the statement about the originality and reliability of our data, as all of our data is taken from published (and in many cases, peer-reviewed) data sources and most of the data, including mitigation costs data, is specific to China. We recognized that this may not have been clear in our first draft, and have substantially expanded the data tables, Methods and Supplementary materials to document our data sources and assumptions. We are not clear on which previous incorrect studies the reviewer is referring to, but we believe our key contributions with our analysis and modeling is not necessarily in correcting previous estimates, but also in:

1. Refining methane emissions for key sectors and quantifying key sources of uncertainties for coal mine methane and rice cultivation
2. Quantifying methane emissions from sources that have not often been reported, including abandoned coal mine methane and aquaculture
3. Updating projections of methane emissions by sources through 2060 under different mitigation scenarios, considering different level of costs of mitigation measures, to understand key areas of methane mitigation potential
4. Quantifying the average mitigation costs for each methane emissions source

We hope the revisions and new information we have added is able to reflect this.

The methodology and data sources haven't been clearly described. Some details of emission data are not provided as the basic information for the mitigation scenarios. Are details available on whether acceptance data were measured (e.g., with a scale) or estimated?

Thank you for this comment. We have added in an expanded Methods section that describes the key input parameters and equations for our modeling calculations, expanded Table 1 to include unabated emission factors, and added two supplementary tables (Table S2 and S3) that includes our activity projection data and describes the data sources for our assumptions of mitigation options and reduction efficiency.

Although all of our data is estimated, not measured, we have compared our estimated emissions for historical years with China's published GHG inventory data for methane to ensure our emission data is well-calibrated (see Table S1).

There needs to be enough information in this paper, either in the main text or supplementary information.

We have added more information to both the main text and supplementary information.

More details about the methodology/analysis would help evaluate appropriateness and reproducibility. The scenarios were defined by the costs prior to 2050 rather than 2060, which needs explanations.

As mentioned above, we have outlined all of our data inputs, assumptions and data sources to help improve evaluation of appropriateness and reproducibility. We have revised the text to clarify that the mitigation measures are defined by current costs based on existing data on China-specific costs, not projected costs out to 2060 that would be highly uncertain and not appropriate for this analysis. Other studies that have analyzed the cost-effectiveness of methane mitigation measures for China and other countries including Teng et al. 2019, EPA 2019, and Höglund-Isaksson 2012 have also based their analysis on currently available cost data, so we believe this methodology is appropriate.

L150, ...destroys or utilizes ventilated air methane (VAM) released from coal mines by 2050, why is 2050? Moreover, this claim is made without sufficient justification.

Thank you for this comment. We have revised this sentence to make the different observations on methane mitigation potential from coal mining in the near-term (before 2050) and later years (after 2050) clearer. Please note that this is not a claim that China will fully adopt VAM mitigation measures by 2050, but it is one of our scenario assumptions about methane mitigation measure adoption. As noted in the Methods section on Scenarios, we assumed that cost-effective methane mitigation measures will be fully deployed by 2050 in the Cost-effective Mitigation Scenario and by 2025 in the Deep Mitigation Scenario. The 2050 endpoint was selected for cost-effective measures because China had previously indicated, and now formally declared prior to COP28, that non-CO2 GHGs will be included in its upcoming NDC for 2035, including carbon neutrality by 2060. For coal mining, the methane mitigation measures will also need to be fully deployed as coal production will need to substantially decrease prior to 2060 for China to achieve its carbon neutrality goal.

L190,...utilizing higher-concentration coal mine methane for power generation in the energy sector, also have negative average abatement costs but a lower reduction potential. It doesn't seem to make sense and needs more explanations.

The difference in mitigation potential is largely due to the assumed deployment of mitigation measures and their methane emissions reduction efficiency based on applicability, costs and technical potential for reducing methane emissions for each of these emission sources: high-concentration coal mine methane,

rice cultivation, aquaculture. As detailed in the new Table S3, utilizing higher-concentration coal mine methane for power generation has 28% reduction efficiency based on the cited reference. In contrast, aquaculture has much higher reduction potential of 100% due to its significantly higher net economic yield resulting from higher stocking intensity of intensive and managed aquaculture systems that also helps reduce methane emissions. We have added a sentence to clarify this key reason for aquaculture.

The inventory developments of different sources and the impacts and effects of cutting emissions during the social transition should be addressed in depth. It is more challenging to address the stochastic and uncertain variables and constraints of the problem than addressing the deterministic ones. It is not clear what each of the mitigation options consist of. There should be more description of each scenario and quantitative information about each scenario, with citation to the source of each quantitative variable or dimension.

Thank you for this comment. We have now added new tables and expanded existing tables to provide more description and quantitative information for each scenario.

For instance, in Additional Measures: Deep Mitigation Scenario, the methane mitigation measures of abandoned coal mines are earlier phasedown due to faster clean energy transition, which is unclear statement, needs more explanations. How were the emission mitigation measures for methane emissions from abandoned mines considered in the scenario setting of the paper? The paper used an abandonment rate of 4.7% for abandoned coal mines in China, how applicable is this parameter to abandoned coal mines in China?

Thanks for this question. We have revised the text in Table 1 to clarify that no direct mitigation measures were considered in the case of abandoned coal mines. However, there is indirect methane mitigation due to different assumptions about China's pace of clean energy transition as coal phasedown translates into less abandoned coal mines in later years. We have added details of our projections for coal reduction and results on abandoned coal mine capacity in the new Table S3.

For the abandonment rate of 4.7%, this was a global rate used in the Kholod et al. 2020 paper. However, other published papers that we cited (including Chen et al. 2022) also analyzed China's abandoned coal mine methane emissions have also applied this rate to China. Please also note that in addition to the 4.7% abandonment rate, our analysis also considers the China-specific endogenously calculated decline in coal production capacity (if any) due to shift away from coal, also consistent with the methodology used in Chen et al. 2022.

The discussion could be strengthened by further interpreting results in relation to previous research, especially the mitigation potential in specific sector. The author didn't give a clear explanation and critical discussion about the policy implications of such study. More intuitively expressions like how to deal with China's CH₄ emissions in the future could be discussed.

Thank you for this comment. We provided a comparison of our estimates of historical methane emissions with other studies but did not do so for the mitigation potential primarily because different studies use different baselines and different definitions for their mitigation scenarios. It would be difficult to compare

such mitigation potential in a meaningful manner when the scenario definitions, assumptions and mitigation costs are different. We agree that the policy implications are important and have added additional discussion on future policy implications in the Discussion section.

Limitations of the study design should be more critically addressed, such as the methodology of projections of key activity driver.

Thank you for this comment. We have added in detailed discussions of our methodology for projecting key activity drivers in the expanded Methods section and also included the raw data for our activity projections in the new Table S2.

The figures are presented roughly, such as Figure 4.

Thank you for this comment. We have revised Figure 4 to improve the formatting for better consistency with other figures.

Proofreading for typos would improve the clarity of writing, especially in L139-144 and 172. The manuscript will benefit from a thorough language check.

Thank you for this comment. We have conducted a thorough language check of the manuscript.

Reviewer #2 (Remarks to the Author):

Summary

The authors present estimates of China’s methane emissions using a bottom-up process, where emissions are a function of economic activities (activity factors) and the estimated emissions intensity of the activities (emissions factors). The authors then present estimates of China’s methane inventories in five-year time steps between 2020 and 2060 under two mitigation scenarios. The authors conclude there are significant opportunities for methane mitigation in China over the period of examination.

Comments

1. Research Question, Relevance and Overall Contribution

Open-access and peer-reviewed information on China’s emissions inventory (including methane) is important, particularly as China does not produce a National Inventory Report for the UNFCCC. NIR methodology is also imperfect, as the UNFCCC methodological guidance is often behind the best-available science. For that reason, this paper potentially offers an important contribution to knowledge. In essence, this is an important avenue of academic research, as it improves understanding and measurement of an important climate-forcing GHG. However, I have major concerns regarding this paper’s relevance and contribution in its current form, in part because of modelling choices and lack of clarity on assumptions.

Thank you for this comment. It is our understanding that China, as a non-Annex I country, is not obligated to produce an annual National Inventory Report for the UNFCCC. As a non-Annex I country, China is obligated to submit national communication (NC) reports and has done so in 2004, 2012, 2019, and most recently in December 2023. Its reporting of GHG emissions inventory lags behind the NC reports, and currently only covers emissions up to 2017 with the latest NC submitted. China’s NC reports are also prepared following the 2006 IPCC guidelines as recommended by the UNFCCC so we agree with your point that both the emissions inventory methodology and the data reported officially by China is outdated. This is one of, but is not the primary goal of our paper as discussed further below.

A. The research question is currently unclear and not explicit. The authors do state what the paper does (“use a bottom-up modeling approach with updated methane emission projections and abatement cost analysis to account for additional sources, uncertainties, and mitigation measures in China’s energy and agricultural sectors”). However, without an explicit question to guide the reader (and why the methods are appropriate to answer it), the authors are not clearly identifying their purpose and contribution to knowledge.

Thank you for this comment. We have made our guiding research questions more explicit in the end of the Introduction, which includes: What are under-reported methane emission sources in China and what are key sources of uncertainties for key emission sources such as coal mine methane? How could China’s methane emission sources change over time through 2060? What are new and emerging opportunities for methane mitigation potential for each emission source and their related costs, given technical feasibility and current cost estimates?

Regarding our contribution to knowledge, we had outlined it in the last section of the Introduction: “First, we account for additional methane emission sources, such as abandoned coal mines and aquaculture, that have not yet been incorporated in multi-sectoral analyses of China. Second, we address uncertainty in methane emissions data in two new ways: by using more granular, region-weighted emission factors for the coal mining sector, and by assessing uncertainties in specific rice cultivation mitigation measures. Third, we evaluate promising new mitigation opportunities, such as biochar application in rice cultivation and aquaculture.”

B. I find it surprising that the authors do not comment on a lack of official Chinese emissions inventory (China does not submit a National Inventory Report to the UNFCCC) as part of the motivation for their work. I am unfamiliar with China’s emissions inventory process (if it exists), and it would be helpful for the authors to discuss this and contrast their inventory approach. This is particularly important given NIR methodology for methane is based on the IPCC fourth assessment report, while science has evolved significantly.

Thanks for this suggestion. We have added a sentence with some background on China’s existing GHG emission inventory efforts and the need for greater research to support improving understanding of existing methane emission sources and potential for future reductions. However, it’s important to emphasize here that our primary research goal is to understand and improve data and uncertainty analysis for individual methane sources, and assess potential reductions in the future given current mitigation measures and their current costs, not just to improve the emissions inventory accounting approach.

C. The authors could and should give a more fulsome discussion of China’s emissions policies. The discussion ignores China’s submissions to the UNFCCC, and couches Chinese policy as responses to US action. Given the goal of the paper is to inform both measurement and mitigation of methane in China, US policy is largely irrelevant whilst a discussion of Chinese policy is important for understanding the context and contribution of the article.

Thank you for this comment. We have revised and reorganized the policy discussion in the Introduction section to provide an overview and update of China’s latest national actions related to methane (e.g., release of the national methane action plan in Nov. 2023 and commitment to include methane in its 2035 NDCs) and introduce its current GHG emissions inventory efforts. We have also added more information on sectoral policies in China that include qualitative mentions of actions that could reduce methane emissions. We have deleted specific mentions of the U.S. action plan.

D. The authors motivate the paper by noting “increasing recognition” that limiting global temperature increases and climate change requires mitigation of GHGs other than carbon dioxide. I find this a false premise; my understanding is that policy focus on carbon dioxide is due to ease of measurement and availability of mitigation options in contrast to other GHGs. Moreover, recent scientific evidence on extensive under-measurement of methane emissions has underscored the imperative nature of addressing it, prompting policy action.

We have reworded and revised the first sentence to be clearer. Our motivation, as emphasized in IPCC's Climate Change 2023 Synthesis Report: A Summary for Policymakers, section B.5 ("From a physical science perspective, limiting human-caused global warming to a specific level requires limiting cumulative CO₂ emissions, reaching at least net zero CO₂ emissions, along with strong reductions in other greenhouse gas emissions. Reaching net zero GHG emissions primarily requires deep reductions in CO₂, methane, and other GHG emissions, and implies net negative CO₂ emissions(39). Carbon dioxide removal (CDR) will be necessary to achieve net negative CO₂ emissions (see B.6). Net zero GHG emissions, if sustained, are projected to result in a gradual decline in global surface temperatures after an earlier peak. (high confidence) {3.1.1, 3.3.1, 3.3.2, 3.3.3, Table 3.1, Cross-Section Box.1}") that reducing CO₂ alone is not sufficient to limit global warming to 1.5C, but that reductions in non-CO₂ GHGs are also needed. We are not saying that this recognition alone is what is driving methane policy focus, but are trying to make the point that there is growing consensus, at least from the scientific community, on the importance of reducing short-lived GHGs such as methane.

2. Assumptions, Methods, and Results

I have major concerns that make me question the paper's relevance and contribution, related to the modelling and underlying assumptions, which I will discuss in detail below.

Without a clear research question and a discussion of China's current inventory methods, alongside clear and transparent analytical methods (all are lacking in the paper as it stands), it is difficult for the reader to understand the contribution. A broad point, regarding exposition, is that the authors do not explicitly describe key features of the model and key assumptions. This is true for both the methane emissions inventory exercise and the mitigation analysis. Accordingly, there is inconsistency in explanation, and these choices makes it nearly impossible to interpret the overall trends and specific results. This means the paper is not particularly accessible, and not replicable.

Thank you for this comment. For the research question, we have outlined our response to the earlier comment 1A above. We have added more background to how China is currently conducting its national GHG emissions inventory and the methods it follows (i.e., 2006 IPCC Guidelines). We have also expanded on our Methods section and added supplementary materials to introduce our modeling features, assumptions and data sources. We hope that with this new information, the paper is more accessible and replicable.

A. The authors state their work "builds on previous work and provides new analysis by using a bottom-up modeling approach with updated assumptions about macroeconomic and physical emissions drivers." The authors fail to describe the exact nature of these updated assumptions and how they differ from extant literature. Table 1 describes at a very high level the activity drivers, the activity projections, and the emissions factors. Figure 1 compares different estimates of China's 2020 methane emissions. However, the authors do not report their actual activity factors or emissions factors or the methodology for determining both. Without this information, it is impossible to evaluate or verify the accuracy of their reported methane emissions. To address this, the authors should include a table with activity factors and emissions factors for each sector, with

sources for each assumption. Ideally, the authors would also justify their assumptions relative to available academic literature and IPCC guidance.

Thank you for this comment. Based on your suggestion, we have added tables with a reporting of our projected activity levels for each methane emission source in Table S1 to complement the methodology and assumptions listed in Table 1 for all activity drivers. We have also added a column to report the unabated emission factors we used for each methane source in Table 1 and added a new details Table S2 to outline our methodology for determining reduction efficiency applied to the emission factor under our two mitigation scenarios and the actual values used in which year, and reported the average abatement costs for each emission source. For all these tables, the data sources and references used are also listed as justification.

B. The authors reference Lin et al. (2019) as the model underlying their results. However, the authors do not even summarise the model approach (beyond stating it is bottom-up). I find this wholly insufficient and strongly urge the authors to, at the very least, outline key equations and assumptions.

Thank you for this comment. We have added in an expanded Methods section that describes the key input parameters and equations for our modeling calculations, expanded Table 1 to include unabated emission factors, and added two supplementary tables (Table S2 and S3) that includes our activity projection data and describes the data sources for our assumptions of mitigation options and reduction efficiency. We have also compared our estimates for 2014 to the latest reported GHG emissions inventory for China in a new Table S1.

C. The authors spend a significant amount of page space on future methane trajectories under three scenarios (reference, cost-effective reductions, and deep mitigation). The underlying assumptions of these scenarios as unclear and undermine the contribution of the paper. Specifically, the reference scenario is “a clean energy transition and decarbonization consistent with Zhou et al. (2022).” This paper is missing from the references and so I was unable to review its assumptions. I do wonder why the authors didn’t just define a “current policy” reference case based on China’s existing policy documents, such as China’s NDC or the Mid-Century Long-Term Low Greenhouse Gas Emission Development Strategy (<https://unfccc.int/documents/307765>). Secondly, the authors define the cost effective mitigation scenario as mitigation measures “below [US] \$10/tCO_{2e} prior to 2050”; what is the rationale for this threshold? Is this real 2023 dollars or a nominal threshold? Similarly, what is the basis for defining deep mitigation as abatement costs below \$100 US/tCO_{2e}?

Thank you for this comment. We included a detailed discussion of the future methane scenarios because we believe this is one of the key contributions of our study in highlighting the techno-economic potential for mitigating methane emissions from different sources given different cost thresholds. We have also added in the missing reference for Zhou et al. 2022. We did define our pace of energy transition based on current policies as outlined in China’s NDC and Mid-Century Strategy, but because China’s methane policies (up to very recently) did not include any quantitative reduction targets, we could not develop a current methane policy scenario. Rather, we based our future scenarios of mitigation potential based on

the technical feasibility and economic costs of individual mitigation measures as applied to China's situation and context, based on the latest literature. This is also why the mitigation measures' costs are based on current costs of these mitigation measures, rather than 2050 or 2060 projected costs.

In terms of the two cost thresholds we used, we set one threshold at US\$10/tCO₂e as low to moderate mitigation costs because this level is similar to the current emissions trading scheme price for carbon in China (see SPG Global, 2023 below for more details). The other higher cost threshold was set at a higher cost level of US\$100/tCO₂e, which is similar to cost thresholds analyzed by other analyses conducted in EPA (2019). The cost is expressed in terms of real 2020 U.S. dollars.

SPG Global, 2023. "China's compliance carbon price hits record high on stronger demand." August 18, 2023. <https://www.spglobal.com/commodityinsights/en/market-insights/latest-news/energy-transition/081823-chinas-compliance-carbon-price-hits-record-high-on-stronger-demand>

D. The authors also fail to source/cite the methane abatement cost estimates. While the authors do note their costs are “[b]ased on a review of several studies of non-CO₂ mitigation measures and abatement costs – i.e., EPA (2019), Höglund-Isaksson (2012), Höglund-Isaksson et al. (2020), and Yang et al. (2014),” this is insufficient detail. Moreover, the authors do not address whether these cost estimates are directly applicable to China. My concern is that a global cost estimate or a country-specific estimate for another country is not directly translatable to a cost estimate for China with different institutions and economic activity.

We have added more documentation on the specific sources for our abatement cost estimates for each methane emission source in Table S3. We have also clarified in the main text that the costs we derived from the cited sources are in fact their estimates or data on China-specific costs so they are directly applicable to China. As noted in greater detail in Table S3, the mitigation costs for many of the newer agricultural mitigation measures (e.g., biochar applications, aquaculture) we analyzed in this paper are the results of meta-analysis of multiple, peer-reviewed Chinese studies based on Chinese case studies and pilots.

E. The authors also discuss uncertainties in methane emissions estimates, but their discussion lacks precision. Specifically, there can be uncertainty in both the activity factors and the emissions factors, and the authors give some examples but do not have a direct comparison of the uncertainty inherent in their approach vis a vis others. Moreover, this uncertainty can come from accuracy (closeness of estimates to the true value) or from precision (range of estimates likely resulting from repeated measurement); the authors do not disclose which source of uncertainty underlies their and others' modelling efforts. This is also relevant for the mitigation scenarios.

Thank you for this comment. We have added more details on the possible sources of uncertainties for activity level data and emissions factor data in bottom-up estimation methods in the Introduction. We have also compared the two key sources of uncertainty that we analyzed (i.e., coal mining EF and rice cultivation mitigation efficacy) with those from the global EDGAR emissions inventory from a new reference, and noting differences in scope of uncertainties considered in our analysis versus the EDGAR emissions inventory. For coal mine methane, only emission factors data was available to assess

uncertainties as coal mining activity data was calibrated to published national statistics. We hope this helps address the concerns noted here.

Minor comments

1. Given the paper's focus on methane, I do not understand why the authors present their results in tonnes of CO₂e.

We have presented our results in CO₂ equivalent terms for two main reasons. First, expressing it in CO₂e terms helps emphasize methane's high global warming potential relative to CO₂, and allows us to compare the different climate impacts given 20-year versus 100-year time frames such as in Figure 4. Second, presenting the results in relative terms to CO₂ is easier for the general scientific audience to understand the scale, as climate change discussions revolve generally around tons of CO₂ emissions, whereas presenting it in physical tons of methane emitted will be hard for a general audience to grasp the relative scale and magnitude.

2. In Figure 1, the authors could make more explicit their estimate (LBNL) relative to the others.

Thank you for this comment. We have added a dashed box around our estimate to make it more explicit, and added a note on this change.

3. Why is the "other" missing in the LBNL estimate in Figure 1? What is included in "other"?

Thanks for this question. We did not have an "other" category as all of our emissions are allocated to the end-use sectors of agriculture, energy and waste as shown. However, other studies and study authors chose to allocate emissions from specific minor sources outside the broader categories of agriculture, energy, and waste, such as specific industrial processes. We have added a note below the chart clarifying the other studies' definitions of the "other" category: "Note: Other category varies by different data sources and includes chemicals, metals and fossil fuel fires in O'Rourke et al. (2021), manufacturing, other transport, chemical and metal industries and fires in Crippa et al. (2021, 2022), stationary and mobile sources in EPA (2019), and average of other estimates from four key sources in IEA (2023)."

4. There are several cross-reference errors in the text.

We apologize for possible Word to PDF conversion issues. We have deleted Word's cross-reference to the figures and tables and manually typed the links to the Figures and Tables to avoid further issues.

5. The paper is poorly formatted, with fragments on text in some places (e.g., line 204 the paragraph starts with the fragment "Figure 6A").

This may also have been due to a formatting conversion error. The sentence should read: "In 2060, the largest source of methane mitigation potential lies in the agriculture sector, which accounts for more than half of the total under both Cost-effective and Deep Mitigation Scenarios (Figure 6A)." We have deleted Word's cross-reference to the figures and tables and manually typed the links to the Figures and Tables to avoid further issues.

This email has been sent through the Springer Nature Tracking System NY-610A-NPG&MTS

REVIEWER COMMENTS

Reviewer #2 (Remarks to the Author):

I appreciate the authors' revisions and responses to my and the other reviewers' comments. I have reviewed the authors' response and the revised paper in detail; in general, I think the paper is improved. However, I still have concerns regarding the relevance and contribution of the paper that were not addressed by the revisions.

1. Research Question, Relevance and Overall Contribution

This is a personal preference, and the authors may choose to disregard this point, but I find the introduction organization non-standard, in that the research question and contribution are discussed at the very end. It is difficult to understand the relevance of the preceding content without knowing the research questions.

Similarly, while I appreciate the authors introducing the three research questions in their revisions, and they use subheadings effectively in the results to link the discussion to the questions, the methods section does not follow the same logic.

I would have also liked to see additional discussion of China's emissions policies; while the authors did add extra content in the introduction, it is lacking specificity (e.g., it's unclear what inclusion of methane mitigation in the 14th Five Year Plan translates into in terms of actionable policy, e.g., targets and regulation to enforce). I raise this because the existing policy environment is important for assessing the reasonableness of the scenario analysis.

2. Assumptions, Methods, and Results

I still find the Methods section lacks transparency and clarity. It is much improved, but does need additional precision. For example, while Table 1 summarises activity projections, items like abandoned coal mine methane are relative to a baseline that is not quantified. It would be helpful to include these data in the supplementary files.

Similarly, assumptions in Table S3 are not always entirely clear, and again specifying baselines would be helpful.

Overall, the paper is still not replicable due to lack of transparency. While the authors have for the most part included emissions factors, they explain activity factors rather than providing the specific parameter values.

More broadly, the authors do not explicitly address my comments on the appropriateness of the scenarios in their revisions. The details in the response letter were not incorporated into substantive in-text revisions. The contribution of this part of the paper rests on the reasonableness of the scenarios (and the underlying assumptions). I would like to see more text justifying the choice of scenarios, as I originally requested.

I also don't quite understand the response re a current policy scenario vs a reference scenario. Is there still insufficient detail on current policy to construct a scenario? Why would you exclude a current policy scenario if there is enough information? A current policy scenario is highly relevant to answering the second research question.

Relatedly, I have reviewed Zhou et al. (2022), and it presents the current state of energy use in China (https://eta-publications.lbl.gov/sites/default/files/ceo_2022_chapter_1_final_2.pdf). Given this, I don't understand how it is a reference/baseline scenario for a clean energy transition without additional explanation.

Reviewer #3 (Remarks to the Author):

Summary:

In this manuscript, the authors conduct a multisectoral analysis of China's methane emissions using a bottom-up approach. They examine methane sources, uncertainties in emissions and emission factors, and opportunities for methane mitigation in China through 2060. They further contextualize their findings under the Cost Effective and Deep Mitigation scenarios. While the work is relevant and fills acknowledged literature gaps, concerns with the methodology still exist. If the authors can address the following comments then the paper would be suitable for publication.

Comments:

The authors discuss benefits and concerns with both bottom-up and top-down approaches. They further mention that the two methods can be used in combination with each other. Despite this

discussion, the authors do not provide a rationale as to why they chose to use a bottom-up approach over a top-down or complementary approach.

The authors make projections through 2060. However, there is no justification provided as to why this year was selected.

In Figure 1, there is no indication as to whether the comparison estimates employed a bottom-up or top-down model. This distinction is important because top-down approaches tend to yield different estimates than bottom-up approaches (Cheewaphongphan et al., 2019).

I find it surprising that the authors do not comment on fugitive methane emissions, which are especially prevalent in oil & gas and coal mining operations. Additionally, it doesn't seem as if natural methane emissions are included in this analysis, despite the presence of several wetlands and lakes across China's geography (Chen et al., 2012, Ding et al., 2004). Alternatively, if this study is only examining anthropogenic methane emissions, then that should be clarified and justified as well.

While the authors describe the equations and the input parameters of their bottom-up model, the purpose and structure of the model are not entirely clear. It would be extremely helpful if the authors were to include a simple diagram showing how exactly the bottom-up spreadsheet works, along with a brief description of the model upon introduction.

Although the authors cite Lin et al., 2019 when describing the equations they used and provide definitions of each variable, there is a lack of clarity as to the theoretical basis of these equations. For example, it would be worthwhile to discuss what emission factors are and how they contribute to the overall equation. A similar explanation on market penetration would be helpful.

The authors may have included the data sources they used, but they fail to mention any biases associated with the data. For example, authors have questioned whether systemic biases may exist in some categories of China's official statistics (Koch-Weser, 2013). Furthermore, while I appreciate that the authors include an analysis of the uncertainties resulting from the data and cite Solazzo et al., 2021 to justify their numbers, some of these uncertainties are very large (e.g., +/-80%). This makes me question the validity of the data sources used. Identification of limitations and biases in the data can possibly explain these results and provide opportunities for other scholars in the field to address these concerns.

The authors touch on coal mining, as a whole. However, methane emissions from specific coal mining methods should be considered because this is a key sector identified by the authors. For example, differences in methane emissions between open cast and shaft mining or artisanal mining and industrial mining should be examined.

Do the authors include informal waste disposal in their calculations? These emissions often go untracked, and therefore, may indicate higher methane emissions than originally estimated. If these emissions were not included, the authors should attempt to incorporate them or state this as a limitation.

There appear to be limitations to the authors' activity projections in table 1. For example, are trends in meat consumption and land-use changes for rice cultivation considered (Yuen et al., 2021)? Additionally, it is likely that the models that the authors cited have limitations that need to be addressed. For example, Kholod et al., 2020, reveal several data gaps in their study, including depth and year of abandonment. I suggest the authors add a "Limitations" column to table 1 to explain these.

In the discussions section of the paper, the role of offset markets is not considered. Offset markets are important when it comes to deploying novel methane removal technologies and reducing costs, particularly in sectors where traditional mitigation measures may be expensive or challenging to deploy (Haya et al., 2020).

Minor Comments:

Line 30: Change "36 time" to "36 times".

Lines 47-50: To improve readability, I suggest changing the sentence to: "At the sectoral level, controls on methane emissions have been qualitatively discussed in other domestic sector plans, such as actions to control and reduce coal capacity, minimize household waste, improve agricultural management, and increase gas recovery and recycling in oil and gas production."

Line 54: Add a comma before 2019.

Lines 72-76: To improve readability, I suggest breaking up the sentence into: "For example, uncertainties arising from bottom-up methods can be attributed to uncertainties in activity-level data. These uncertainties stem from factors such as the incompleteness or lack of representativeness of statistical sampling, as well as the imputation of missing data and

extrapolation for future years. Additionally, uncertainties in emission factors (EFs) may arise due to the representativeness of a limited number of observations, inaccuracies in assumptions and/or source aggregation, as well as biases, variability, and/or random errors (Solazzo et al., 2021).”

Lines 385-386: “higher-cost” should be two separate words.

The authors only consider crop straw biochar. However, they may also consider discussing wood biochar, which is cheaper than straw biochar (Shackley et al., 2011).

Do the authors include imported embodied methane emissions in their analysis of China’s national methane emissions? For example, methane emissions are generated when China imports primary materials from Australia (Fan et al., 2024). If these imported emissions are not already included, their omission should at least be discussed.

I noticed the authors did not include a mitigation measure for abandoned coal mine methane in Table S3. However, solutions, such as abandoned mine methane extraction via drilling and suction (UNECE, 2019), do exist, and should be included if possible.

Some sources are not cited (e.g. Jiang et al., 2021, Zhou et al., 2020).

Citations to the sources used in this peer review:

Cheewaphongphan, P., Chatani, S., & Saigusa, N. (2019). Exploring Gaps between Bottom-Up and Top-Down Emission Estimates Based on Uncertainties in Multiple Emission Inventories: A Case Study on CH₄ Emissions in China. In *Sustainability* (Vol. 11, Issue 7, p. 2054). MDPI AG.
<https://doi.org/10.3390/su11072054>

Chen, H., Zhu, Q., Peng, C., Wu, N., Wang, Y., Fang, X., Jiang, H., Xiang, W., Chang, J., Deng, X., & Yu, G. (2012). Methane emissions from rice paddies natural wetlands, lakes in China: synthesis new estimate. In *Global Change Biology* (Vol. 19, Issue 1, pp. 19–32). Wiley.
<https://doi.org/10.1111/gcb.12034>

Ding, W., Cai, Z., & Wang, D. (2004). Preliminary budget of methane emissions from natural wetlands in China. In *Atmospheric Environment* (Vol. 38, Issue 5, pp. 751–759). Elsevier BV.
<https://doi.org/10.1016/j.atmosenv.2003.10.016>

Fan, Z., Ju, X., Tong, H., Liang, Z., Sun, N., Mao, H., & Peng, J. (2024). Environmental impacts of potential mining-replacing-import alternative for China in response to the China-Australia coal ban. In *Journal of Cleaner Production* (Vol. 442, p. 140876). Elsevier BV. <https://doi.org/10.1016/j.jclepro.2024.140876>

Haya, B., Cullenward, D., Strong, A. L., Grubert, E., Heilmayr, R., Sivas, D. A., & Wara, M. (2020). Managing uncertainty in carbon offsets: insights from California's standardized approach. In *Climate Policy* (Vol. 20, Issue 9, pp. 1112–1126). Informa UK Limited. <https://doi.org/10.1080/14693062.2020.1781035>

Koch-Weser, I. (2013). The Reliability of China's Economic Data: An Analysis of National Output. <https://www.uscc.gov/sites/default/files/Research/TheReliabilityofChina'sEconomicData.pdf>

Shackley, S., Hammond, J., Gaunt, J., & Ibarrola, R. (2011). The feasibility and costs of biochar deployment in the UK. In *Carbon Management* (Vol. 2, Issue 3, pp. 335–356). Informa UK Limited. <https://doi.org/10.4155/cmt.11.22>

UNECE (2019). Best Practice Guidance for Effective Methane Recovery and Use from Abandoned Coal Mines ECE ENERGY SERIES No. 64. https://globalmethane.org/documents/Best_Practice_Guidance_for_Effective_Methane_Recovery_and_Use_from_Abandoned_Coal_Mines_FINAL__with_covers_.pdf

Yuen, K. W., Hanh, T. T., Quynh, V. D., Switzer, A. D., Teng, P., & Lee, J. S. H. (2021). Interacting effects of land-use change and natural hazards on rice agriculture in the Mekong and Red River deltas in Vietnam. In *Natural Hazards and Earth System Sciences* (Vol. 21, Issue 5, pp. 1473–1493). Copernicus GmbH. <https://doi.org/10.5194/nhess-21-1473-2021>

China's Methane Mitigation Potential: An Assessment of Costs and Uncertainties through 2060

Nina Khanna¹, Jiang Lin^{1,2*}, Xu Liu³, and Wenjun Wang²

Affiliations: ¹Energy Technologies Area, Lawrence Berkeley National Laboratory; Berkeley, CA 94720, USA; ²University of California at Berkeley; Berkeley, CA 94720, USA; ³Peking University, Beijing, China.

*Corresponding author. Email: j_lin@lbl.gov

Responses to Reviewer Comments (second round of review):

Responses to reviewer comments are shown in blue font below.

REVIEWER COMMENTS

Reviewer #2 (Remarks to the Author):

I appreciate the authors' revisions and responses to my and the other reviewers' comments. I have reviewed the authors' response and the revised paper in detail; in general, I think the paper is improved. However, I still have concerns regarding the relevance and contribution of the paper that were not addressed by the revisions.

1. Research Question, Relevance and Overall Contribution

This is a personal preference, and the authors may choose to disregard this point, but I find the introduction organization non-standard, in that the research question and contribution are discussed at the very end. It is difficult to understand the relevance of the preceding content without knowing the research questions.

Thank you for this comment. We have reviewed several recent *Nature Communications* articles to observe the organization of their Introductions. Based on those that we reviewed and your comment, we have restructured the Introduction to introduce the research questions directly following the summary of literature reviewed on certain topics, rather than present all three at the end. We have also followed other recent *Nature Communication* articles in ending with a paragraph that summarizes our methodological contributions and relevance of our analysis.

Similarly, while I appreciate the authors introducing the three research questions in their revisions, and they use subheadings effectively in the results to link the discussion to the questions, the methods section does not follow the same logic.

Thank you for this helpful feedback. We have made the following changes to the Methods section:

- added headers to better distinguish the three key subsections of the Methods section,
- rewritten the Methods section to include a high-level introduction of bottom-up models with the theoretical equations and a visual diagram of the key modeling framework with inputs and outputs;
- distinguished the discussion of data sources and assumptions for three key inputs of activity drivers, emission factors, and costs; and
- revised the heading and text to clarify the section on methane emission and mitigation scenarios.

We hope this helps improve the logic of this section.

I would have also liked to see additional discussion of China's emissions policies; while the authors did add extra content in the introduction, it is lacking specificity (e.g., it's unclear what inclusion of methane mitigation in the 14th Five Year Plan translates into in terms of actionable policy, e.g., targets and regulation to enforce). I raise this because the existing policy environment is important for assessing the reasonableness of the scenario analysis.

Thank you for this comment. We have added a more detailed description of the 14th Five Year Plan mention of methane and the recently released National Methane Action Plan in the Introduction: **“Domestically, China’s 14th Five-Year Plan for national economic development, endorsed in 2021, explicitly included text that China will “strengthen the control” of non-CO₂ GHGs including methane for the first time.” And “The national methane action plan prioritizes significantly enhancing methane monitoring, reporting and verification systems, and calls for effectively improving methane utilization, emissions control technologies and policy frameworks in the energy, agriculture and waste sectors (MEE, 2023a). However, the action plan did not set quantitative targets on methane emissions control, and only included four quantitative goals for increasing the utilization of coal mine gas, reutilization of livestock waste and urban household waste, and harmless disposal of urban sludge.”** As noted in the new text, China's current methane-related policies do not include any quantitative goals for reducing aggregate (national total) or sectoral methane emissions reductions, and the four quantitative goals related to methane emission sources do not translate directly into methane reductions (with the exception of coal mine gas utilization which we already included in our calibration of the base year). As a result of the lack of data on actionable policies for reducing methane emissions, we did not include methane mitigation measures in our baseline, Reference Scenario. We have also added this rationale to the Scenarios discussion section in Methods.

More background information and analysis on the lack of quantitative actions or goals in China's current methane-related policies can be found at: <https://www.carbonbrief.org/qa-what-does-chinas-new-methane-plan-mean-for-its-climate-goals/>

2. Assumptions, Methods, and Results

I still find the Methods section lacks transparency and clarity. It is much improved, but does need additional precision. For example, while Table 1 summarises activity projections, items like

abandoned coal mine methane are relative to a baseline that is not quantified. It would be helpful to include these data in the supplementary files.

Thank you for this comment. As noted above, we have added sub-section headers and new text to help improve the transparency and clarity of this section and its organization. We previously provided the activity projections for each subsector in Table S2 but realized that the descriptions may be a bit confusing in Table 1. We have added notes in Table 1 to highlight where activity levels may differ between the scenarios for the energy sector as a result of two different underlying energy transitions modeled.

Similarly, assumptions in Table S3 are not always entirely clear, and again specifying baselines would be helpful.

Thank you for this note. We have added a footnote under Table S3 noting that the baseline Reference Scenario does not consider any methane mitigation measures. We also added text to the scenario descriptions in the Introduction and Methods section to note this.

Overall, the paper is still not replicable due to lack of transparency. While the authors have for the most part included emissions factors, they explain activity factors rather than providing the specific parameter values.

Thank you for raising this concern, but we provided the specific parameter values for all activity projections in Table S2.

More broadly, the authors do not explicitly address my comments on the appropriateness of the scenarios in their revisions. The details in the response letter were not incorporated into substantive in-text revisions. The contribution of this part of the paper rests on the reasonableness of the scenarios (and the underlying assumptions). I would like to see more text justifying the choice of scenarios, as I originally requested.

Thank you for this helpful comment. We have expanded the descriptions of the scenarios in the Methods section to clarify the different baselines used (e.g., gradual vs. accelerated energy transition, no methane mitigation vs. methane mitigation), and added rationale on our two cost thresholds for the two methane mitigation scenarios. We have also added text to clarify how the cost-analysis we conducted links with our choices for assumptions for our two mitigation scenarios.

I also don't quite understand the response re a current policy scenario vs a reference scenario. Is there still insufficient detail on current policy to construct a scenario? Why would you exclude a current policy scenario if there is enough information? A current policy scenario is highly relevant to answering the second research question.

For the energy sectors, there is sufficient information – particularly quantitative targets for specific fuels and subsectors – to represent current policies in our baseline scenario. For the Reference Scenario and Cost-Effective Mitigation Scenario, we essentially adopted this “current policy”-consistent scenario for the energy sectors. For the Deep Mitigation Scenario, we adopted

an accelerated clean energy transition scenario beyond current policy goals, based on the work outlined in Zhou et al. 2022.

For the non-energy sectors and for methane specifically, all of the recent policies in China do not include any direct quantitative targets for methane reduction beyond a coal mine gas utilization target for 2025. These policies mention many different actions that could be adopted to reduce methane emissions, but do not include explicit goals or guidance on the pace or scale of adopting such mitigation actions. As a result, there is insufficient information to construct a reasonable “current policy” scenario for methane mitigation. For this reason, we use no methane mitigation as the baseline and define two plausible pathways of methane mitigation based on the marginal abatement cost thresholds of US\$10/tCO₂e and US\$100/tCO₂e. We have added text to clarify this in the scenario descriptions.

Relatedly, I have reviewed Zhou et al. (2022), and it presents the current state of energy use in China (https://eta-publications.lbl.gov/sites/default/files/ceo_2022_chapter_1_final_2.pdf). Given this, I don't understand how it is a reference/baseline scenario for a clean energy transition without additional explanation.

We apologize if the incorrect link was shared last time. The scenarios for clean energy transition are presented in Chapter 3 of this full report: <https://international.lbl.gov/sites/default/files/2022-04/China%20Energy%20Outlook%202022-full%20report%2004.22.22.pdf>

As noted in Chapter 3 of that report, the two energy scenarios modeled are a baseline scenario (called “Continued Improvement Scenario”) that is intended to serve as a reference scenario of what China *could* achieve if it fully adopts the maximum feasible shares of today's commercially available, cost-effective energy efficiency and renewable energy supply by 2050. It considers continuation of all existing policies and meeting of all announced policy goals and targets, but also assumes that additional policies will be introduced to support the full adoption of today's commercially available and cost-effective technologies for a clean energy transition.

The alternative scenario of faster clean energy transition (called “Deep Mitigation Scenario”) assumes that China will fully develop and deploy deep decarbonization technologies, practices, and some behavioral changes, including technologies that are currently in pilot or demonstration phases, to reduce GHG emissions as much as technically feasible by 2050, without adopting carbon capture, use, and storage (CCUS) at a large scale. This scenario includes multiple strategies for energy efficiency improvement and other energy demand reductions, accelerated electrification, fuel switching and technological upgrades (including new, alternative fuels), and broader structural changes that impact energy-related activities.

Please note neither of these scenarios include non-CO₂ GHGs, and the projections are only up to 2050. For 2050 to 2060, we extended energy projections based on other comparable modeling scenarios from Jiang et al. 2021 and Yu et al. 2022 that model energy transitions through 2060.

Reviewer #3 (Remarks to the Author):

Summary:

In this manuscript, the authors conduct a multisectoral analysis of China's methane emissions using a bottom-up approach. They examine methane sources, uncertainties in emissions and emission factors, and opportunities for methane mitigation in China through 2060. They further contextualize their findings under the Cost Effective and Deep Mitigation scenarios. While the work is relevant and fills acknowledged literature gaps, concerns with the methodology still exist. If the authors can address the following comments then the paper would be suitable for publication.

Comments:

The authors discuss benefits and concerns with both bottom-up and top-down approaches. They further mention that the two methods can be used in combination with each other. Despite this discussion, the authors do not provide a rationale as to why they chose to use a bottom-up approach over a top-down or complementary approach.

Thank you for this comment and question. We have chosen to use a bottom-up approach because of data availability, a standardized methodology that has been used in IPCC proceedings and adopted globally by countries for reporting their GHG emissions, and applicability for greater granularity in sectoral emissions analysis and for projecting future emissions. While top-down approaches based on measurements of methane concentrations (e.g., through airborne, satellite, or on-road sensors) can help improve site-specific estimates of methane emissions and record large emitting events that result from accidents or unpredictable process failures such as those in oil and gas operations, there is currently insufficient measurement data to provide estimates for multiple sectors and sources for China. In addition, top-down approaches are not directly applicable for projecting future emissions. To explain this, we have added new text to the manuscript:

“For methane emissions specifically, top-down measurement initiatives have only recently emerged to help improve measurement and reporting of current and historical emissions, but these assessments remain incomplete and most countries – including China – have little or no measurement-based data (IEA 2023).” And **“In the absence of robust measurement-based data for multiple sectors for China, this study uses a standardized bottom-up method based on existing international guidelines for greenhouse gas emission inventories to estimate the historical and to project China's future methane emissions”** in the beginning of the Results section.

The authors make projections through 2060. However, there is no justification provided as to why this year was selected.

Thank you for this question. We have added text to the Introduction to explain that 2060 was used as the end-year for this analysis because it is the target year for China's carbon neutrality goal.

In Figure 1, there is no indication as to whether the comparison estimates employed a bottom-up

or top-down model. This distinction is important because top-down approaches tend to yield different estimates than bottom-up approaches (Cheewaphongphan et al., 2019).

Thank you for this question. We have reviewed the studies cited and confirmed that with the exception of IEA 2023, all of the cited emissions data are based on bottom-up inventory-based estimation approaches. IEA's 2023 Global Methane Tracker data takes a hybrid approach where it attempts to incorporate findings from satellite-based campaigns and large emission events detected by satellites, as well as peer-reviewed measurement-based and top-down estimates in literature at a global level. We did not include purely top-down estimates in our comparison as we used a bottom-up approach for estimating historical emissions, and it would not be directly comparable to top-down estimates for the reasons mentioned in Cheewaphongphan et al., 2019. We have added new text to explain that the comparison is focused only on bottom-up or inventory approaches and noted the exception for IEA's methodological approach.

I find it surprising that the authors do not comment on fugitive methane emissions, which are especially prevalent in oil & gas and coal mining operations. Additionally, it doesn't seem as if natural methane emissions are included in this analysis, despite the presence of several wetlands and lakes across China's geography (Chen et al., 2012, Ding et al., 2004). Alternatively, if this study is only examining anthropogenic methane emissions, then that should be clarified and justified as well.

Thank you for this comment. The coal mine methane, oil and natural gas methane emissions we modeled and analyzed are all fugitive and vented emissions. We used other terms such as "leakage" in Table S3 to describe fugitive methane emissions, but have added "fugitive" to the table text as well. You are correct in that our scope is only focused on anthropogenic methane emissions. While we appreciate the natural methane emissions, we chose to focus our analysis only on anthropogenic methane emissions because that is the scope covered under China's methane-related policies such as the National Methane Action Plan, and we have added text in the Introduction, Results and Methods section to clarify this.

While the authors describe the equations and the input parameters of their bottom-up model, the purpose and structure of the model are not entirely clear. It would be extremely helpful if the authors were to include a simple diagram showing how exactly the bottom-up spreadsheet works, along with a brief description of the model upon introduction.

Thank you for this helpful suggestion. We have more text describing the model and have added the diagram below showing the emission sources modeled, inputs and outputs of the bottom-up model.

**Note*T&D is transmission and distribution. Applicable mitigation measures are selected based on scenario cost thresholds and the China-specific marginal abatement cost of individual mitigation measure.*

Although the authors cite Lin et al., 2019 when describing the equations they used and provide definitions of each variable, there is a lack of clarity as to the theoretical basis of these equations. For example, it would be worthwhile to discuss what emission factors are and how they contribute to the overall equation. A similar explanation on market penetration would be helpful.

Thank you for this comment and question. We have added text to 1) clarify that the theoretical basis for the equations is actually based on the internationally-accepted IPCC guidelines for bottom-up greenhouse gas emission inventories and the mitigation assessment based on U.S. Environmental Protection Agency's approach, and 2) explain in greater detail how emission factors and market adoption (revised from market penetration) contribute to the overall equation for calculating unabated methane emissions and methane mitigation. The new text in Methods is as follows:

“This study uses a bottom-up modeling approach to analyze in detail the anthropogenic methane emission sources in energy and non-energy sectors from a techno-economic perspective and to evaluate future pathways to lower methane emissions. In bottom-up modeling, specific methane-emitting actions and mitigation technologies are modeled at the level of specific emission sources, such as coal mining or enteric fermentation. Methane emissions are calculated as a product of the emitting activity source and a source-specific emission factor that could change with the market adoption of mitigation measure or technology. The modeling methodology described below is directly adapted from the methodology used by the IPCC and the U.S. EPA in their respective Guidelines for Greenhouse Gas Emission Inventories (IPCC 2006, EPA 2019).”

The authors may have included the data sources they used, but they fail to mention any biases associated with the data. For example, authors have questioned whether systemic biases may

exist in some categories of China's official statistics (Koch-Weser, 2013). Furthermore, while I appreciate that the authors include an analysis of the uncertainties resulting from the data and cite Solazzo et al., 2021 to justify their numbers, some of these uncertainties are very large (e.g., +/-80%). This makes me question the validity of the data sources used. Identification of limitations and biases in the data can possibly explain these results and provide opportunities for other scholars in the field to address these concerns.

Thank you for this comment. In reviewing Koch-Weser 2013, we noted that the analysis was focused primarily on statistical weaknesses in China's economic data, namely GDP and its components such as consumption and investment, fixed asset investment, etc. We also reviewed another article (Plekhanov 2017) that reviewed 88 publications critiquing Chinese official statistics, and found that most researchers focused their analysis on three main economic indicators of GDP, unemployment and industrial production. However, neither article provided clear and citable information on systemic biases or uncertainties in China's official statistics, particularly those such as non-economic indicators used in this study. In the absence of such information or verifiable alternative data sources, we are not able to adjust the statistical data utilized in our analysis. Based on this, we have added a note in the Data sources subsection of the Methods section noting that despite ongoing progress to improve its statistical work, there are still issues with China's official statistics system, on which the historical activity levels from this study are largely based and cited the two articles. Overall, however, we believe that our estimates for 2020 methane emissions is still valid because it is comparable and in the range of 6 other studies that used different data sources and assumptions. The new text added is:

“Most of the historical activity data series were derived from official national statistics, which have been critiqued for deficiencies in methodology and transparency with examples of data inconsistency and discrepancies, particularly for economic indicators (Koch-Weser 2013, Plekhanov 2017). However, in the absence of alternative data that can be verified, we followed the official statistics as they are also the basis for China's UNFCCC inventory reporting and our estimates for 2020 methane emissions are in-line with other independent estimates as seen in Figure 1.”

New Reference cited:

Plekhanov D., 2017. Quality of China's Official Statistics: A Brief Review of Academic Perspectives. *The Copenhagen Journal of Asian Studies* 35 (1): 76-101.

The authors touch on coal mining, as a whole. However, methane emissions from specific coal mining methods should be considered because this is a key sector identified by the authors. For example, differences in methane emissions between open cast and shaft mining or artisanal mining and industrial mining should be examined.

Thank you for this comment. We did consider differences in methane emissions from open cast and underground shaft mining, but made the simplifying assumption that the relative shares of production from these two different methods will remain constant over time (as noted in Table 1) as the scope of our paper was focused on a multi-sector analysis. However, the emission factor we calculated from Sheng et al. 2019 does account for differences in methane emissions from

these two types of mining methods as it is based on actual mine-specific reported emissions from the three largest provinces, which includes two provinces (Inner Mongolia and Shanxi) that accounts for most of the open-pit mining in China (Liu et al. 2023).

Similarly, we did not explicitly model any potential differences between artisanal vs. industrial mining for coal for two primary reasons: 1) artisanal coal mining (e.g., mines with less than 0.3 Mt/year of mining capacity) accounts for a very small share (4.5%) of production based on Liu et al. 2023, and 2) China has focused on consolidating and closure of smaller coal mines due to concerns with safety, environmental impacts and inefficient operations since 2012 so artisanal coal mining will continue to decline.

New Reference cited in response:

Liu X., Li L, and Y. Yang. 2023. "Development status of coal mining in China." *Journal of the Southern African Institute of Mining and Metallurgy* 123 (1). <http://dx.doi.org/10.17159/2411-9717/1506/2023>

Do the authors include informal waste disposal in their calculations? These emissions often go untracked, and therefore, may indicate higher methane emissions than originally estimated. If these emissions were not included, the authors should attempt to incorporate them or state this as a limitation.

Thank you for this question. We did not include informal waste disposal in our analysis, as it is our understanding that it has become an increasingly small share of total municipal solid waste (MSW) and due to significant data limitations. Based on Kurniawan et al. 2022, uncontrolled waste disposal only accounted for 1.8% of total MSW in 2020. This is largely because the Chinese government launched reforms to significantly modernize MSW management after 2018 with its Solid Waste law, and set a target requiring compulsory waste sorting in 46 major Chinese cities by end of 2020. These cities have since established formal waste collection and sorting systems for household waste, which was formally a key source of waste for the informal recycling sector. Local officials are also closing down informal waste and recycling networks, suggesting that their role will continue to be diminished. A recent analysis focused on analyzing carbon flows in China's MSW also could not account for the informal sector due to data limitations on quantity and composition of recyclables informally picked out of waste streams (Zhang et al. 2024). We have noted this as a limitation in our study.

Additional references cited in response:

Kurniawan, T., Liang X., O'Callaghan E., et al. Transformation of Solid Waste Management in China: Moving towards Sustainability through Digitalization-Based Circular Economy. *Sustainability* 14 (4): 2374 (2022).

Zhang J., Du H., Wang T., et al. Tracking the carbon flows in municipal waste management in China. *Scientific Reports* 14, 1471 (2024).

There appear to be limitations to the authors' activity projections in table 1. For example, are

trends in meat consumption and land-use changes for rice cultivation considered (Yuen et al., 2021)? Additionally, it is likely that the models that the authors cited have limitations that need to be addressed. For example, Kholod et al., 2020, reveal several data gaps in their study, including depth and year of abandonment. I suggest the authors add a "Limitations" column to table 1 to explain these.

Thank you for this suggestion. We have added two new sections in the text discussing our methodology and assumptions for projecting activity drivers to discuss limitations with the activity drivers. We believe incorporating the explanations directly to the text can help shed light on the two different types of limitations. The new text we added are:

“For abandoned coal mine, the global rate of abandonment used by Kholod et al. 2020 and subsequent China-focused studies (Sheng et al. 2021, Chen et al. 2022) is also used due to persisting data gaps in China-specific abandoned mine depths, timing of abandonment, and decay curves.”

And

“While historical trends and insights from sectoral experts are incorporated into the future activity projections, the projections for non-energy activity drivers notably do not include significant paradigm shifts that may be needed for deep decarbonization due to limitations from uncertainty on timing and scalability of these shifts. For example, dietary changes, food waste reduction and other lifestyle changes could significantly affect future activity drivers such as growth trends in livestock population, rice and aquaculture demand, and waste diversion trends, but these are not considered in our current model.”

In the discussions section of the paper, the role of offset markets is not considered. Offset markets are important when it comes to deploying novel methane removal technologies and reducing costs, particularly in sectors where traditional mitigation measures may be expensive or challenging to deploy (Haya et al., 2020).

Thank you for this note. We briefly mentioned a role for certified carbon credits in the Discussions section but based on your suggestion, have expanded the text on role for offset markets for both coal mine methane and agriculture methane reductions and have cited Haya et al. 2020. The new text in the Discussions section is shown below in bold.

“In addition, providing financial incentives through certified carbon credits **under China’s voluntary carbon market for China Certified Emission Reduction credits** or other green financial measures can help accelerate the adoption of methane mitigation measures with moderate costs such as VAM mitigation. **The emissions reduction impact of these certified carbon credits can be strengthened by offset protocols developed to provide a standardized approach for reducing the risk of over-crediting emissions reductions from mine methane capture, such as the one developed in California in 2014 (Haya et al. 2020).**”

and

Long-term government financing as well as potentially increased private investments through inclusion of agricultural projects in China’s re-started voluntary carbon market can provide the research and development funding needed to increase commercial viability and deployment.

Minor Comments:

Line 30: Change “36 time” to “36 times”.

Thank you for the suggestion, we have made this change.

Lines 47-50: To improve readability, I suggest changing the sentence to: “At the sectoral level, controls on methane emissions have been qualitatively discussed in other domestic sector plans, such as actions to control and reduce coal capacity, minimize household waste, improve agricultural management, and increase gas recovery and recycling in oil and gas production.”

Thank you for the helpful suggested edit. We have adopted the changes to this sentence.

Line 54: Add a comma before 2019.

Thank you for this suggestion, we have made this change.

Lines 72-76: To improve readability, I suggest breaking up the sentence into: “For example, uncertainties arising from bottom-up methods can be attributed to uncertainties in activity-level data. These uncertainties stem from factors such as the incompleteness or lack of representativeness of statistical sampling, as well as the imputation of missing data and extrapolation for future years. Additionally, uncertainties in emission factors (EFs) may arise due to the representativeness of a limited number of observations, inaccuracies in assumptions and/or source aggregation, as well as biases, variability, and/or random errors (Solazzo et al., 2021).”

Thank you very much for these helpful edits to improve clarity and readability. We have adopted your edited text directly.

Lines 385-386: “higher-cost” should be two separate words.

Thank you for this suggestion. We have changed all instances of “higher-cost” to “higher cost.”

The authors only consider crop straw biochar. However, they may also consider discussing wood biochar, which is cheaper than straw biochar (Shackley et al., 2011).

Thank you for this suggestion. We focused on straw biochar because crop residues are the most prevalent form of biochar feedstock in China. According to a recent study by Xia et al. 2023, “Distribution of biomass waste varies across regions, and at the national level, crop residues (48.8%) and livestock manure (25.4%) and food waste (15.6%) are the main sources of biochar feedstocks and should be prioritized in biochar development. Forest residues have a marginal contribution due to competing uses from other sectors, and sewage sludge derived biochar is a net source of GHG emissions due to the large energy consumption required for sludge drying.”

As a result, we have decided not to add wood-based biochar in our analysis.

New reference cited in response: Xia L., Chen W., Lu B. et al. 2023. "Climate mitigation potential of sustainable biochar production in China." *Renewable and Sustainable Energy Reviews* 175, 113145. <https://doi.org/10.1016/j.rser.2023.113145>

Do the authors include imported embodied methane emissions in their analysis of China's national methane emissions? For example, methane emissions are generated when China imports primary materials from Australia (Fan et al., 2024). If these imported emissions are not already included, their omission should at least be discussed.

We follow the IPCC Guidelines for national GHG Emissions Inventories in our methodology for estimating and projecting methane emissions, which limits the scope of emissions accounting to "greenhouse gas emissions and removals taking place within national territory and offshore area over which the country has jurisdiction" (IPCC, 2006 and IPCC, 2019) and do not include embodied emissions of imported goods. This follows a production-based, rather than consumption-based, approach to reporting emissions. Under the IPCC conventions, the embodied emissions of imported products would not be included in national reporting of emissions, as the embodied emissions of exported products (e.g., embodied methane emissions of meat exported from China to other countries) would also not be excluded from national emissions. The Fang et al. 2024 article evaluates the change in GHG emissions and pollutants from increased domestic coal mining as a result of the China-Australia coal ban, which would implicitly be included in the scope of our analysis as the change occurs in domestic production. However, the methane emissions associated with mining of coal in Australia would not be in the scope of our analysis.

Additional references included in response:

Intergovernmental Panel on Climate Change (IPCC), 2006. 2006 IPCC Guidelines for National Greenhouse Gas Inventories, Prepared by the National Greenhouse Gas Inventories Programme, Eggleston H.S., Buendia L., Miwa K., Ngara T. and Tanabe K. (eds). Published: IGES, Japan.

IPCC, 2019. 2019 Refinements to the 2006 IPCC Guidelines for National Greenhouse Gas Inventories, Chapter 8: Reporting Guidance and Tables. <https://www.ipcc.ch/report/2019-refinement-to-the-2006-ipcc-guidelines-for-national-greenhouse-gas-inventories/>

I noticed the authors did not include a mitigation measure for abandoned coal mine methane in Table S3. However, solutions, such as abandoned mine methane extraction via drilling and suction (UNECE, 2019), do exist, and should be included if possible.

Thank for this note. We did not consider mitigation measures for abandoned coal mine methane because the Chen et al. 2022 study we adapted our AMM analysis from did not include any data or information, particularly related to costs or scalability, for AMM mitigation. Based on your comment, we found a more recent study (Kang et al. 2023) that also concluded it was not technically feasible to conduct large-scale mitigation for abandoned coal mines, and that AMM gas utilization is still in the experimental stage due to complex geological conditions and

immature technologies, and that pilot projects are needed to increase awareness and demonstration. This is also consistent with the UNECE report, which notes that favorable mining, geological and market conditions are all needed for AMM mitigation projects. To note these limitations, we have added a note on why direct mitigation measures are not considered for AMM in Table 2 and Table S3.

New Reference cited:

Kang Y., Tian P., Li J. et al. 2023. Methane mitigation potentials and related costs of China's coal mines. *Fundamental Research*, In Press. <https://doi.org/10.1016/j.fmre.2023.09.012>

Some sources are not cited (e.g. Jiang et al., 2021, Zhou et al., 2020).

Thank you for catching these two omissions. We have added a full reference for Jiang et al. 2021 and replaced Zhou et al. 2020 with an updated citation for the Zhou et al. 2022 final report that is now publicly available.

Jiang K., He C., Jiang W. et al. 2021. "Transition of the Chinese Economy in the Face of Deep Greenhouse Gas Emissions Cuts in the Future." *Asian Economic Policy Review* 16 (1): 142-162.

Zhou N., Khanna N., Zhang J. et al. 2022. "China Energy Outlook 2022." Berkeley, California: Lawrence Berkeley National Laboratory. LBNL-2001444. <https://international.lbl.gov/sites/default/files/2022-04/China%20Energy%20Outlook%202022-full%20report%2004.22.22.pdf>

Citations to the sources used in this peer review:

Cheewaphongphan, P., Chatani, S., & Saigusa, N. (2019). Exploring Gaps between Bottom-Up and Top-Down Emission Estimates Based on Uncertainties in Multiple Emission Inventories: A Case Study on CH₄ Emissions in China. In *Sustainability* (Vol. 11, Issue 7, p. 2054). MDPI AG. <https://doi.org/10.3390/su11072054>

Chen, H., Zhu, Q., Peng, C., Wu, N., Wang, Y., Fang, X., Jiang, H., Xiang, W., Chang, J., Deng, X., & Yu, G. (2012). Methane emissions from rice paddies natural wetlands, lakes in China: synthesis new estimate. In *Global Change Biology* (Vol. 19, Issue 1, pp. 19–32). Wiley. <https://doi.org/10.1111/gcb.12034>

Ding, W., Cai, Z., & Wang, D. (2004). Preliminary budget of methane emissions from natural wetlands in China. In *Atmospheric Environment* (Vol. 38, Issue 5, pp. 751–759). Elsevier BV. <https://doi.org/10.1016/j.atmosenv.2003.10.016>

Fan, Z., Ju, X., Tong, H., Liang, Z., Sun, N., Mao, H., & Peng, J. (2024). Environmental impacts of potential mining-replacing-import alternative for China in response to the China-Australia coal ban. In *Journal of Cleaner Production* (Vol. 442, p. 140876). Elsevier BV.

<https://doi.org/10.1016/j.jclepro.2024.140876>

Haya, B., Cullenward, D., Strong, A. L., Grubert, E., Heilmayr, R., Sivas, D. A., & Wara, M. (2020). Managing uncertainty in carbon offsets: insights from California's standardized approach. In *Climate Policy* (Vol. 20, Issue 9, pp. 1112–1126). Informa UK Limited. <https://doi.org/10.1080/14693062.2020.1781035>

Koch-Weser, I. (2013). The Reliability of China's Economic Data: An Analysis of National Output. <https://www.uscc.gov/sites/default/files/Research/TheReliabilityofChina'sEconomicData.pdf>

Shackley, S., Hammond, J., Gaunt, J., & Ibarrola, R. (2011). The feasibility and costs of biochar deployment in the UK. In *Carbon Management* (Vol. 2, Issue 3, pp. 335–356). Informa UK Limited. <https://doi.org/10.4155/cmt.11.22>

UNECE (2019). Best Practice Guidance for Effective Methane Recovery and Use from Abandoned Coal Mines ECE ENERGY SERIES No. 64. https://globalmethane.org/documents/Best_Practice_Guidance_for_Effective_Methane_Recovery_and_Use_from_Abandoned_Coal_Mines_FINAL_with_covers_.pdf

Yuen, K. W., Hanh, T. T., Quynh, V. D., Switzer, A. D., Teng, P., & Lee, J. S. H. (2021). Interacting effects of land-use change and natural hazards on rice agriculture in the Mekong and Red River deltas in Vietnam. In *Natural Hazards and Earth System Sciences* (Vol. 21, Issue 5, pp. 1473–1493). Copernicus GmbH. <https://doi.org/10.5194/nhess-21-1473-2021>

** See Nature Portfolio's author and referees' website at www.nature.com/authors for information about policies, services and author benefits.

This email has been sent through the Springer Nature Tracking System NY-610A-NPG&MTS

Confidentiality Statement:

This e-mail is confidential and subject to copyright. Any unauthorised use or disclosure of its contents is prohibited. If you have received this email in error please notify our Manuscript Tracking System Helpdesk team at <http://platformsupport.nature.com> .

Details of the confidentiality and pre-publicity policy may be found here <http://www.nature.com/authors/policies/confidentiality.html>

Privacy Policy | Update Profile

REVIEWER COMMENTS

Reviewer #2 (Remarks to the Author):

I want to compliment the authors on their revisions; the paper is much improved.

I have only a few remaining minor comments:

1. In Figure 2c, the Reference Scenario does not have uncertainty bars. I find this surprising, given the earlier in-text discussion of the uncertainty for rice cultivation (-40% to +70% for EFs). Is this an oversight? Regardless, it should be addressed in a figure note or in text. This is perhaps explained in lines 183 through 187, but it is not entirely clear.
2. To be consistent across figures and ensure clarity, please use panels a and b in describing Figure 3; split Figure 4 into four panels; split Figure 6 into three panels.
3. All figures are missing source notes.
4. Figure 5 and the in-text discussion: currency units are not clear, e.g., is this nominal 2030 dollars or current dollars discounted from expected 2030 costs.
5. Please also ensure currency units are clear, including USD and short versus metric tonnes, real versus nominal etc. If USD, is there a yuan-to-USD conversion rate used?

China's Methane Mitigation Potential: An Assessment of Costs and Uncertainties through 2060

Nina Khanna¹, Jiang Lin^{1,2*}, Xu Liu³, and Wenjun Wang²

Affiliations: ¹ Energy Technologies Area, Lawrence Berkeley National Laboratory; Berkeley, CA 94720, USA; ² University of California at Berkeley; Berkeley, CA 94720, USA; ³ Peking University, Beijing, China.

*Corresponding author. Email: j_lin@lbl.gov

Responses to reviewer **in bold**

REVIEWER COMMENTS

Reviewer #2 (Remarks to the Author):

I want to compliment the authors on their revisions; the paper is much improved.

Thank you for this compliment. We are glad we have successfully addressed previous review comments.

I have only a few remaining minor comments:

1. In Figure 2c, the Reference Scenario does not have uncertainty bars. I find this surprising, given the earlier in-text discussion of the uncertainty for rice cultivation (-40% to +70% for EFs). Is this an oversight? Regardless, it should be addressed in a figure note or in text. This is perhaps explained in lines 183 through 187, but it is not entirely clear.

Yes, as we attempted to note in lines 183 through 187, we did not include the inherent uncertainty in rice cultivation emission factor (-40% to +70% for EFs) in global emission inventories using Tier 1 methodology as we used a China-specific emission factor. However, we did include and evaluate the significantly lower uncertainty range of $\pm 10\%$ identified for rice cultivation mitigation efficiency for biochar efficacy specifically for the two mitigation scenarios. This $\pm 10\%$ uncertainty is not shown in the Reference Scenario as biochar is not applied as a mitigation measure in the Reference Scenario. We have added a note in the revised legend to clarify this.

2. To be consistent across figures and ensure clarity, please use panels a and b in describing Figure 3; split Figure 4 into four panels; split Figure 6 into three panels.

Thank you for this suggestion. We have made these revisions and updated the figure titles and legends.

3. All figures are missing source notes.

For Fig 1, we cited the data sources for the other studies' estimates. The other figures are all based on our analysis and we have added a note that the source data is provided.

4. Figure 5 and the in-text discussion: currency units are not clear, e.g., is this nominal 2030 dollars or current dollars discounted from expected 2030 costs.

Thank you for this question. We have added clarification of the currency unit (2020 nominal US dollars) in the main text and in the legend of Figure 5.

5. Please also ensure currency units are clear, including USD and short versus metric tonnes, real versus nominal etc. If USD, is there a yuan-to-USD conversion rate used?

Thank you for this note. We have noted the yuan-to-USD conversion rate for 2020 nominal dollars in Figure 5 legend. We had noted that all emissions are expressed in metric tonnes in line 135 but have changed previous uses of "tons" in Table 1 to "tonne" to be consistent.

** See Nature Portfolio's author and referees' website at www.nature.com/authors for information about policies, services and author benefits.

This email has been sent through the Springer Nature Tracking System NY-610A-NPG&MTS